# BIG LEARNING: A UNIVERSAL MACHINE LEARNING PARADIGM?

## ABSTRACT

Recent breakthroughs based on big/foundation models reveal a vague avenue for AI, that is, *big data, big/foundation models, big learning,* · · · . Following that avenue, here we elaborate on our newly introduced big learning. Specifically, big learning exhaustively exploits the information/tasks inherent in its large-scale *complete/incomplete* training data, by learning to simultaneously model many/all joint/conditional/marginal data distributions (thus named big learning) with one universal foundation model. We reveal that big learning is what existing foundation models are implicitly doing; accordingly, our big learning provides high-level guidance for flexible design and improvements of foundation models. Besides, big learning ($i$) is equipped with great flexibilities for complete/incomplete training data and for customizing trustworthy data tasks; ($ii$) potentially delivers all joint/conditional/marginal data capabilities after training; ($iii$) significantly reduces the training-test gap with improved model generalization; and ($iv$) potentially unifies conventional machine learning paradigms and enables their flexible cooperations, manifested as a universal learning paradigm. Preliminary experiments verified the effectiveness of the presented big learning.

## 1 INTRODUCTION

AI is undergoing a paradigm shift with the rise of big/foundation models (Bommasani et al., 2021; Yuan et al., 2022), *e.g.,* BERT (Stickland & Murray, 2019), GPT-3 (Brown et al., 2020), DALL-Es (Ramesh et al., 2021; 2022), MAE (He et al., 2021), *etc.* Foundation models, often based on mask-and-predict pretraining and downstream finetuning, are capable of benefiting from pretraining on broad data at scale and accordingly, demonstrate diverse downstream task capabilities with impressive robustness (Stickland & Murray, 2019), adaptability (He et al., 2021), and generalization (Ramesh et al., 2021). Therefore, they are rapidly being integrated into real-world AI systems, *e.g.,* BERT into Google search, Codex (Chen et al., 2021a) into GitHub's Copilot, *etc.*

Despite the impressive capabilities and characteristics of foundation models, a unified theoretical framework justifying their successes remains missing (Bommasani et al., 2021; Yuan et al., 2022), which is crucial for their further improvements and is likely a milestone for the foundation model community (Tamkin et al., 2021).

To address that challenge, we first notice that the successes of foundation models are mainly attributed to the following two properties, in addition to increasingly powerful parallel computing techniques.

- **Data comprehensiveness.** Foundation models are not picky about their training data and therefore embrace flexible training on massive easily-accessible data with great diversity (*e.g.,* those crawled from the Internet). These training data, thanks to their massiveness, diversity, and minimal human interventions in the collection, are likely more consistent with the "true" data distribution that underlies both training and test phases, leading to a narrowed training-test gap *from the data perspective* and serving as one reason for the improved generalization and robustness of foundation models.

- **Task comprehensiveness.** A foundation model is often pretrained in a massive-task manner on a wealth of *data tasks* (like mask-and-predict), which can be flexibly specified as modeling some conditional data distributions across potentially diverse domains (see Section 3 for details). Such massive-task and potentially diverse pretraining of foundation models narrows the pretraining-finetuning/training-test gap *from the learning perspective*, *i.e.,* it's likely the downstream task

resembles a pretraining one, and therefore contributes to the successes of foundation models. Moreover, the massive-task pretraining may encourage learning compositional intrinsic meta-knowledge encoded in the model parameters (Lu et al., 2021; Aghajanyan et al., 2021), which may hold the key for out-of-distribution generalization (Bommasani et al., 2021).

Based on the above observations and by reviewing the development history of deep learning, we perceive a vague avenue for AI, that is *big data, big/foundation models, big learning,* ⋯⋯. Specifically, one leverages *big data* to comprehensively represent the underlying data distribution, develops *big/foundation models* to serve as a big information "container," relies on *big learning* to comprehensively and exhaustively convey data information into that container, and so on. Accordingly, different from existing machine learning paradigms that only exploit limited information contained in training data, we present big learning for exhaustive data information exploitation, following that AI avenue.

The presented big learning further strengthens the above-mentioned data and task comprehensiveness by leveraging a universal foundation model to simultaneously model *many/all* joint/conditional/marginal data distributions (across potentially diverse domains), manifested as a "big" training task that exhaustively exploits the data information. Such big learning behavior closely resembles the fundamental unconscious mind and the vision system of human brains, which are excellent at comprehensive information exploitation in a multitasking manner (Bargh & Morsella, 2008; Mesquita, 2015; Ludwig et al., 2014; Saarela & Landy, 2015).

Our big learning comes with three main contributions.

- It serves as a theoretical platform for analyzing, justifying, and improving big/foundation models, because most of them are implicitly doing (parts of) big learning, as revealed in Section 3.

- By modeling *many/all* joint/conditional/marginal data distributions, big learning ($i$) comprehensively exploits the available data information and embraces statistical sharing power to encourage summarizing intrinsic compositional meta-knowledge within model parameters and ($ii$) potentially delivers all joint/conditional/marginal data capabilities after training, which are of great value *e.g.,* for arbitrary data completion, flexible counter-factual analysis, and reasoning.

- It delivers extraordinary data and training-task flexibilities by enabling large-scale training with complete/incomplete data on diverse learning tasks across different domains, leading to ($i$) minimal human interventions in data collection and learning-task specification, ($ii$) significantly reduced training-test (or pretraining-finetuning) gap, and ($iii$) potentially a universal machine learning paradigm that unifies and enables cooperations among conventional ones.

## 2  RELATED WORK AND PRELIMINARY

**Big/Foundation models.** Taking shape in NLP, big/foundation models have drastically changed the research and practice of AI (Bommasani et al., 2021; Yuan et al., 2022). BERT (Stickland & Murray, 2019) and GPT series (Radford et al., 2019; Brown et al., 2020) significantly accelerate the development of natural language processing, while models like DALL-Es (Ramesh et al., 2021; 2022) effectively promote interdisciplinary research among different research fields. Most foundation models are pretrained in a mask-and-predict manner, *i.e.,* holding out a portion of the input followed by training the model to use the remaining parts to predict that held-out portion. We will reveal in Section 3 that such mask-and-predict pretraining is a special case of the proposed big learning, which accordingly reveals the underlying principle of foundation models and serves as a theoretical platform for their analysis, justification, and further improvements.

**Transformers and Vision Transformers (ViTs).** Based on the self-attention mechanism (Vaswani et al., 2017), Transformers have been serving as the de facto model architecture for foundation models. Often Transformers (like BERT) take as input a sequence of discrete indexes $\boldsymbol{x} \in \mathbb{Z}^L$ with length $L$ and output the corresponding latent embedding $\boldsymbol{h} \in \mathbb{R}^{L \times D}$ with embedding dimension $D$ for downstream applications; attentions are implemented among the $L$ locations layer-wisely. ViTs (Dosovitskiy et al., 2020) are Transformers modified for dealing with continuous images, which have been empirically proven to have better generalization and robustness than convolutional neural networks (Naseer et al., 2021). Different from Transformers embedding discrete indexes into high-dimensional continuous features, ViTs directly employ flattened image patches as those features, as demonstrated in Fig. 1b. It's well known that Transformers/ViTs are over-parameterized Lan et al.

(2019) and therefore data/information hungry; we will reveal that this property of Transformers/ViTs, together with their great modeling flexibility, exactly matches our big learning.

**Multi-mode learning objectives.** Two well-known multi-mode learning objectives are ($i$) the cross-entropy loss, often used in maximum likelihood learning with *discrete* categorical observations, and ($ii$) the GAN loss (Goodfellow et al., 2014) for adversarial learning on *continuous* observations.

- Given feature-label/history-current-word pairs $(\boldsymbol{x}, y) \sim q(\boldsymbol{x}, y), y \in \{1, \cdots, C\}$ and a model $p_{\boldsymbol{\theta}}(y|\boldsymbol{x})$ modeling the categorical distributed $y$ given $\boldsymbol{x}$, the cross-entropy loss is identical to

$$\text{KL}[q(\boldsymbol{x}, y)||p_{\boldsymbol{\theta}}(y|\boldsymbol{x})q(\boldsymbol{x})] \propto \mathbb{E}_{q(\boldsymbol{x})}\mathbb{E}_{q(y|\boldsymbol{x})}[-\log p_{\boldsymbol{\theta}}(y|\boldsymbol{x})], \tag{1}$$

where the optimal $p_{\boldsymbol{\theta}^*}(y|\boldsymbol{x}) = q(y|\boldsymbol{x})$. Note the categorical distribution is capable of modeling multiple modes, *e.g.,* consider the diverse generation from the GPT-3 (Brown et al., 2020).

- Generative adversarial nets (GANs) are widely used for synthesizing highly realistic images (Karras et al., 2019a;b; 2021). A standard GAN (Goodfellow et al., 2014) consists of a generator $G_{\boldsymbol{\theta}}$ and a discriminator $D_{\boldsymbol{\phi}}$, both of which are trained in an adversarial manner via

$$\min_{\boldsymbol{\theta}} \max_{\boldsymbol{\phi}} \ \mathbb{E}_{\boldsymbol{x} \sim q(\boldsymbol{x})} \log D_{\boldsymbol{\phi}}(\boldsymbol{x}) + \mathbb{E}_{p_{\boldsymbol{\theta}}(\boldsymbol{x})} \log(1 - D_{\boldsymbol{\phi}}(\boldsymbol{x})), \tag{2}$$

where $q(\boldsymbol{x})$ is the underlying data distribution and $p_{\boldsymbol{\theta}}(\boldsymbol{x})$ is the generated distribution with the generative process $\boldsymbol{x} = G_{\boldsymbol{\theta}}(\boldsymbol{z}), \boldsymbol{z} \sim p(\boldsymbol{z})$. $p(\boldsymbol{z})$ is an easy-to-sample distribution, like a normal distribution. With optimal $D_{\boldsymbol{\phi}^*}$, Eq. (2) minimizes the Jensen-Shannon divergence $\text{JS}[q(\boldsymbol{x})||p_{\boldsymbol{\theta}}(\boldsymbol{x})]$ (Goodfellow et al., 2014). Recently, the community begins to exploit integrating ViTs into GANs to benefit from their modeling flexibility (Jiang et al., 2021; Lee et al., 2021; Zhao et al., 2021; Zhang et al., 2021). We also employ the ViT-based GAN generator and discriminator in the experiments, so as to leverage ViTs' modeling flexibility and over-parameterization property to meet the modeling requirement of our big learning.

## 3 BIG LEARNING: A UNIVERSAL MACHINE LEARNING PARADIGM

For better introduction of our big learning, we first present its main idea in simplified unsupervised settings, where a data sample $\boldsymbol{X} = (\boldsymbol{x})$ contains only a feature $\boldsymbol{x} \in \mathbb{R}^{L \times D}$ (with length $L$ and dimension $D$, like $L$ flattened patches of an image or $L$ words with $D = 1$), followed by generalizing its scope to the general settings with a data sample $\boldsymbol{X} = (\boldsymbol{y}, \boldsymbol{x})$ containing an additional supervision $\boldsymbol{y} \in \mathbb{R}^{L^{\boldsymbol{y}} \times D^{\boldsymbol{y}}}$ (*e.g.,* when $L^{\boldsymbol{y}} = D^{\boldsymbol{y}} = 1, y \in \{1, \cdots, C\}$ may represent a label). Note in both cases, our big learning can naturally handle "incomplete data," which are defined as either $\boldsymbol{x}$ missing values along the $L$-dimension (like missing image patches) or $\boldsymbol{y}$ missing values along the $L^{\boldsymbol{y}}$-dimension.

### 3.1 UNSUPERVISED BIG LEARNING

In unsupervised settings, we focus on generation tasks for the introduction. Given a collection of data samples $\{\boldsymbol{x}_1, \boldsymbol{x}_2, \cdots\}$ from the underlying data distribution $q(\boldsymbol{x})$, the mainstream machine learning paradigms concentrate solely on the joint modeling, *i.e.,* to construct a joint model $p_{\boldsymbol{\theta}}(\boldsymbol{x})$ to resemble the joint data distribution $q(\boldsymbol{x})$, or informally $p_{\boldsymbol{\theta}}(\boldsymbol{x}) \longrightarrow q(\boldsymbol{x})$, using GANs (Brock et al., 2019; Karras et al., 2019a), VAEs (Kingma & Welling, 2013; Dai & Wipf, 2019), Flows (Dinh et al., 2014; Kingma & Dhariwal, 2018), diffusion models (Ho et al., 2020; Song et al., 2020), *etc.*

**Motivations.** We highlight two practical situations where the joint modeling is restricted.

1. For most practical applications like those in medical/biological scenarios, only a limited portion of the accessible data are complete and can be utilized by joint modeling. It's inexpedient to simply discard all incomplete ones (and the valuable information therein), especially where the data collection is expensive. Moreover, discarding incomplete data likely introduces unexpected interventions that violate the *i.i.d.* assumption, which lays the foundation of deep learning.

2. It's worth highlighting that, given a dataset with complete data, one already receives the data samples from all joint/conditional/marginal distributions; therefore, ideally, one should comprehensively exploit that valuable information *e.g.,* to form all the associated data capabilities (like vari-

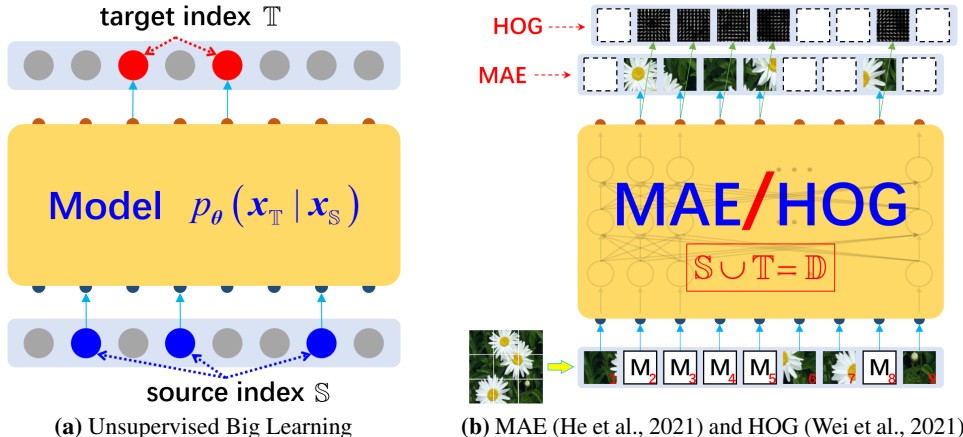

**(a)** Unsupervised Big Learning      **(b)** MAE (He et al., 2021) and HOG (Wei et al., 2021)

Figure 1: Unsupervised big learning (a) and its special cases (b). Often a mask token `[M]` is inserted to the input locations outside $\mathbb{S}$ for forward propagation, while no loss is back-propagated to the output locations outside $\mathbb{T}$. Note inserting the `[M]` tokens later in a middle layer (but at the same location) often lightens the computation and memory burdens but improves the performance (He et al., 2021).

ous conditional sampling for data completion) or to leverage different joint/conditional/marginal perspectives (formed as different training tasks) to regularize each other[1].

The above analyses motivate us to model all joint/conditional/marginal distributions simultaneously (manifested as *"big" learning with massive tasks*), to enable flexible training with all available complete/incomplete data[2] and, at the same time, comprehensively "collect" data capabilities via exhaustive data exploitation. Ideally, to collect all joint/conditional/marginal data capabilities, one need construct $N_{\text{all}} = \sum_{i=0}^{L-1} C_L^i (\sum_{k=1}^{L-i} C_{L-i}^k)^3$ models in total (See Appendix A for details), which is clearly prohibitive. Alternatively, we propose the following unsupervised big learning.

**Unsupervised big learning.** For the unsupervised settings with $\boldsymbol{x} \in \mathbb{R}^{L \times D}$ and the index set $\mathbb{L} = \{1, \cdots, L\}$, the proposed unsupervised big learning leverages a universal foundation model $p_{\boldsymbol{\theta}}(\boldsymbol{x}_{\mathbb{T}}|\boldsymbol{x}_{\mathbb{S}})$ to model many/all joint/conditional/marginal data distributions simultaneously, *i.e.*,

$$p_{\boldsymbol{\theta}}(\boldsymbol{x}_{\mathbb{T}}|\boldsymbol{x}_{\mathbb{S}}) \longrightarrow q(\boldsymbol{x}_{\mathbb{T}}|\boldsymbol{x}_{\mathbb{S}}), \tag{3}$$

where $\mathbb{S} \subset \mathbb{L}$ and $\mathbb{T} \subseteq \mathbb{L}, \mathbb{T} \neq \emptyset$ denote any *non-overlapping* source/target index sets, respectively. $q(\boldsymbol{x}_{\mathbb{T}}|\boldsymbol{x}_{\mathbb{S}})$ is the underlying conditional data distribution, whose samples are readily selected from training data. Note $\mathbb{S} \cup \mathbb{T}$ need not be $\mathbb{L}$, indicating that unsupervised big learning can naturally handle incomplete data (with the model architecture and training objective detailed below). Fig. 1 demonstrates unsupervised big learning and Table 1 compares it with the conventional joint modeling.

Table 1: Comparing joint modeling with unsupervised big learning.

| Compared Methods | Joint Modeling | Unsupervised Big Learning |
|---|---|---|
| Intuitively | Straight-forward | Complicated/Intractable |
| Training Data | Complete Data | Complete/Incomplete Data |
| Data Information Exploitation | Single Joint Perspective | Exhaustive Many/All Perspectives |
| Capabilities After Training | Joint | Joint/Conditional/Marginal |
| Potential Downstream Applications | Limited | Extremely Abundant |

**Model architecture and training objective.** Since the lengths of source $\boldsymbol{x}_{\mathbb{S}}$ and target $\boldsymbol{x}_{\mathbb{T}}$ are not fixed, it's not easy to model the universal $p_{\boldsymbol{\theta}}(\boldsymbol{x}_{\mathbb{T}}|\boldsymbol{x}_{\mathbb{S}})$ based on convolutions. Motivated by the modeling flexibility of Transformers/ViTs and the fact that most foundation models are built on top of them, we propose to model the universal $p_{\boldsymbol{\theta}}(\boldsymbol{x}_{\mathbb{T}}|\boldsymbol{x}_{\mathbb{S}})$ based on the transformer architecture.

---

[1]If the joint modeling is learned perfectly, it's possible but often computationally expensive to recover all conditional/marginal capabilities. However, that perfect modeling assumption is usually violated in practice.

[2]Incomplete data are readily exploited in the corresponding conditional/marginal tasks.

[3]$C_L^i$ denotes the number of $i$-combinations from a set with $L$ elements. Note in this paper, we only consider the joint/conditional/marginal distributions *w.r.t.* the length $L$, *e.g.*, consider the NLP setup with $D = 1$.

The training objective is data-specific and can be specified as commonly-used machine learning objectives, as exampled below, where we reveal that common foundation models are implicitly doing big learning.

1. Let $\boldsymbol{x}$ denotes a sequence of *continuous* features, such as flattened patches of an image (He et al., 2021), as demonstrated in Fig. 1b. Unsupervised big learning aims at acquiring the data capabilities of generating a subset of image patches $\boldsymbol{x}_{\mathbb{T}}$ given another subset $\boldsymbol{x}_{\mathbb{S}}$, manifested as versatile data completions ($\mathbb{S} \neq \emptyset$) or joint/marginal generations ($\mathbb{S} = \emptyset$). Considering the varying lengths of $\boldsymbol{x}_{\mathbb{S}}/\boldsymbol{x}_{\mathbb{T}}$, we resort to the attention mechanism to handle that challenge and, accordingly, construct $p_{\boldsymbol{\theta}}(\boldsymbol{x}_{\mathbb{T}}|\boldsymbol{x}_{\mathbb{S}})$ as a modified ViT, which models the generative process of $\boldsymbol{x}_{\mathbb{T}}$ conditioned on $\boldsymbol{x}_{\mathbb{S}}$, similar to a (conditional-)GAN generator. For practical applications, $\boldsymbol{x}_{\mathbb{T}}$ are usually multi-mode; therefore, the training objective need have the multi-mode capability. We propose to use the GAN loss as detailed in the following Eqs. (4) and (5), where we again leverage a ViT-based GAN discriminator to handle varying-length $\boldsymbol{x}_{\mathbb{S}}/\boldsymbol{x}_{\mathbb{T}}$.

   One can of course consider other multi-mode extensions (such as maximum likelihood learning via *e.g.,* VAEs, Flows, diffusion models, and EBMs (LeCun, 2022)) and even the simple unimodal Gaussian case as in the MAE (He et al., 2021). Specifically, the MAE employs $p_{\boldsymbol{\theta}}(\boldsymbol{x}_{\mathbb{T}}|\boldsymbol{x}_{\mathbb{S}}) = \mathcal{N}(\boldsymbol{x}_{\mathbb{T}}|\boldsymbol{\mu}_{\boldsymbol{\theta}}(\boldsymbol{x}_{\mathbb{S}}), \mathbf{I})$, where $\boldsymbol{\mu}_{\boldsymbol{\theta}}(\boldsymbol{x}_{\mathbb{S}})$ is a modified ViT, $\mathbb{S}$ is a 25% subset of $\mathbb{L}$, and $\mathbb{T} = \mathbb{L}\backslash\mathbb{S}$. It's thus clear that the MAE is a special case of our unsupervised big learning.

2. Let $\boldsymbol{x}$ denotes a sequence of *discrete* tokens, like text words or vector-quantified image patches (Ramesh et al., 2021). Unsupervised big learning predicts the target/masked tokens $\boldsymbol{x}_{\mathbb{T}}$ given the source ones $\boldsymbol{x}_{\mathbb{S}}$ for many/all $(\mathbb{S}, \mathbb{T})$ pairs. To model the correlations among the target tokens $\boldsymbol{x}_{\mathbb{T}}$, we propose to construct the universal $p_{\boldsymbol{\theta}}(\boldsymbol{x}_{\mathbb{T}}|\boldsymbol{x}_{\mathbb{S}})$ as a conditional language model, which autoregressively generates $\boldsymbol{x}_{\mathbb{T}}$ (*w.r.t.* the order specified by $\mathbb{T}$) conditioned on the bidirectionally extracted $\boldsymbol{x}_{\mathbb{S}}$-features. Therefore, thanks to the random ordering in $\mathbb{T}$, unsupervised big learning will deliver versatile generation/completion *w.r.t.* any predicting order[4] after training.

   The unsupervised big learning is closely related to the permutation language modeling (PLM) proposed in the XLNet (Yang et al., 2019), except that PLM only considers the conditional modeling with $\mathbb{S} \cup \mathbb{T} = \mathbb{L}$ and a fixed 85%/15% split for $\mathbb{S}/\mathbb{T}$; it's therefore clear that the XLNet implicitly implements (a special case of) unsupervised big learning. Similarly, the masked language modeling of the BERT can be recovered by further specifying conditional independent assumption among $\boldsymbol{x}_{\mathbb{T}}$ tokens. With $\mathbb{S} = \emptyset$ and the only forward autoregressive order for $\mathbb{T}$, unsupervised big learning readily reduces to the causal language modeling of GPTs.

**Take the former *continuous* settings for an example.** For simplicity, we illustrate with the standard GAN loss (Goodfellow et al., 2014). Given a universal model $p_{\boldsymbol{\theta}}(\boldsymbol{x}_{\mathbb{T}}|\boldsymbol{x}_{\mathbb{S}})$ that models the generative processes of $\boldsymbol{x}_{\mathbb{T}}$ given $\boldsymbol{x}_{\mathbb{S}}$ for all $(\mathbb{S}, \mathbb{T})$ pairs, one can

1. match any model distribution $p_{\boldsymbol{\theta}}(\boldsymbol{x}_{\mathbb{T}}|\boldsymbol{x}_{\mathbb{S}})q(\boldsymbol{x}_{\mathbb{S}})$ to the corresponding underlying (subset) data distribution $q(\boldsymbol{x}_{\mathbb{S}\cup\mathbb{T}})$ with

$$\min_{\boldsymbol{\theta}} \max_{\boldsymbol{\phi}} \mathbb{E}_{q(\boldsymbol{x}_{\mathbb{S}\cup\mathbb{T}})} \log \sigma[f_{\boldsymbol{\phi}}(\boldsymbol{x}; \mathbb{S}, \mathbb{T})] + \mathbb{E}_{p_{\boldsymbol{\theta}}(\boldsymbol{x}_{\mathbb{T}}|\boldsymbol{x}_{\mathbb{S}})q(\boldsymbol{x}_{\mathbb{S}})} \log \sigma[-f_{\boldsymbol{\phi}}(\boldsymbol{x}; \mathbb{S}, \mathbb{T})], \quad (4)$$

   where the optimal $f_{\boldsymbol{\phi}^*}(\boldsymbol{x}; \mathbb{S}, \mathbb{T}) = \log \frac{q(\boldsymbol{x}_{\mathbb{S}\cup\mathbb{T}})}{p_{\boldsymbol{\theta}}(\boldsymbol{x}_{\mathbb{T}}|\boldsymbol{x}_{\mathbb{S}})q(\boldsymbol{x}_{\mathbb{S}})} = \log \frac{q(\boldsymbol{x}_{\mathbb{T}}|\boldsymbol{x}_{\mathbb{S}})}{p_{\boldsymbol{\theta}}(\boldsymbol{x}_{\mathbb{T}}|\boldsymbol{x}_{\mathbb{S}})}$. To handle the varying lengths of $\mathbb{S}/\mathbb{T}$, $f_{\boldsymbol{\phi}}(\boldsymbol{x}; \mathbb{S}, \mathbb{T})$ is also constructed as a modified ViT; see Appendix C for details.

2. enable "communications" among any two model distributions with $\mathbb{S}^1 \cup \mathbb{T}^1 = \mathbb{S}^2 \cup \mathbb{T}^2$ via

$$\min_{\boldsymbol{\theta}} \max_{\boldsymbol{\phi}} \begin{cases} \mathbb{E}_{p_{\boldsymbol{\theta}}(\boldsymbol{x}_{\mathbb{T}^1}|\boldsymbol{x}_{\mathbb{S}^1})q(\boldsymbol{x}_{\mathbb{S}^1})} \log \sigma[f_{\boldsymbol{\phi}}(\boldsymbol{x}; \mathbb{S}^2, \mathbb{T}^2) - f_{\boldsymbol{\phi}}(\boldsymbol{x}; \mathbb{S}^1, \mathbb{T}^1)] \\ + \mathbb{E}_{p_{\boldsymbol{\theta}}(\boldsymbol{x}_{\mathbb{T}^2}|\boldsymbol{x}_{\mathbb{S}^2})q(\boldsymbol{x}_{\mathbb{S}^2})} \log \sigma[f_{\boldsymbol{\phi}}(\boldsymbol{x}; \mathbb{S}^1, \mathbb{T}^1) - f_{\boldsymbol{\phi}}(\boldsymbol{x}; \mathbb{S}^2, \mathbb{T}^2)], \end{cases} \quad (5)$$

   where the "communication" discriminator can be implicitly constructed with the same neural network $f_{\boldsymbol{\phi}}(\boldsymbol{x}; \mathbb{S}, \mathbb{T})$ from Eq. (4). Proofs are given in Appendix B.

Note $(\mathbb{S}, \mathbb{T})$ can be flexibly sampled from all possible pairs (or a predefined subset) according to the actual situations of the problem of interest. How to "optimally" specify the pretraining settings for $(\mathbb{S}, \mathbb{T})$ is beyond the scope of this paper. Here, we focus on demonstrating the feasibility of (unsupervised) big learning, *i.e.,* one can train a universal foundation model to yield many/all joint/conditional/marginal data capabilities, informally $p_{\boldsymbol{\theta}^*}(\boldsymbol{x}_{\mathbb{T}}|\boldsymbol{x}_{\mathbb{S}}) = q(\boldsymbol{x}_{\mathbb{T}}|\boldsymbol{x}_{\mathbb{S}})$ for all $(\mathbb{S}, \mathbb{T})$ pairs.

---

[4]The previous continuous example need not consider the predicting order, thanks to its (conditionally) joint modeling and generation of $\boldsymbol{x}_{\mathbb{T}}$ via GANs.

### 3.2 DISCUSSIONS ON UNSUPERVISED BIG LEARNING

The following discussions are readily extended to our big learning presented in Section 3.3.

**Can we share one universal foundation model $p_{\boldsymbol{\theta}}(\boldsymbol{x}_{\mathbb{T}}|\boldsymbol{x}_{\mathbb{S}})$ among all $(\mathbb{S}, \mathbb{T})$ pairs? Yes, and it's what we should do.** It's clear that all conditional/marginal data distributions $q(\boldsymbol{x}_{\mathbb{T}}|\boldsymbol{x}_{\mathbb{S}})$ can be derived from the joint one $q(\boldsymbol{x})$, meaning that their perfect modelings should share the same set of parameters. Being consistent with that fact, (unsupervised) big learning employs the universal $p_{\boldsymbol{\theta}}(\boldsymbol{x}_{\mathbb{T}}|\boldsymbol{x}_{\mathbb{S}}), \forall (\mathbb{S}, \mathbb{T})$ with shared parameters $\boldsymbol{\theta}$ to simultaneously model all joint/conditional/marginal data distributions. The great successes from existing foundation models (like BERT, XLNet, MAE, *etc.*) have extensively verified the effectiveness of that universal modeling of (unsupervised) big learning, under certain special settings with $\mathbb{S} \cup \mathbb{T} = \mathbb{L}$, fixed $\mathbb{S}/\mathbb{T}$ ratio, but various $(\mathbb{S}, \mathbb{T})$ pairs.

**On the model capacity of $p_{\boldsymbol{\theta}}(\boldsymbol{x}_{\mathbb{T}}|\boldsymbol{x}_{\mathbb{S}})$.** To collect many/all data capabilities within one universal foundation model $p_{\boldsymbol{\theta}}(\boldsymbol{x}_{\mathbb{T}}|\boldsymbol{x}_{\mathbb{S}})$ brings tremendous challenges to its model capacity. Fortunately, Transformers/ViTs are well-known to be data/information hungry, along with their modeling flexibility and parallel-computing amenability, making them well suited to model $p_{\boldsymbol{\theta}}(\boldsymbol{x}_{\mathbb{T}}|\boldsymbol{x}_{\mathbb{S}})$. Moreover, huge Transformers are emerging, *e.g.,* the BaGuaLu with $174$ trillion parameters (Ma et al., 2022). Therefore, the model capacity is likely not an issue for (unsupervised) big learning (see the experiments).

**Generalization to diverse domains.** Considering practical situations, to directly model within the observed domain, *i.e.,* $p_{\boldsymbol{\theta}}(\boldsymbol{x}_{\mathbb{T}}|\boldsymbol{x}_{\mathbb{S}})$, may not be a good choice (He et al., 2021; Wei et al., 2021). We reveal that, with trustworthy domain knowledge, one may alternatively do (unsupervised) big learning in diverse transformed domains, *e.g.,* via (i) $p_{\boldsymbol{\theta}}(\hat{\boldsymbol{x}}_{\mathbb{T}}|\hat{\boldsymbol{x}}_{\mathbb{S}})$ with $\hat{\boldsymbol{x}} = g(\boldsymbol{x})$ or (ii) $p_{\boldsymbol{\theta}}(h(\boldsymbol{x}_{\mathbb{T}})|k(\boldsymbol{x}_{\mathbb{S}}))$ as exampled in Fig. 1b, where $g(\cdot)$, $h(\cdot)$, and $k(\cdot)$ are domain-knowledge-inspired functions.

**On the generalization of model parameters and latent features.** As aforementioned, exiting big/foundation models, showing extraordinary robustness, adaptability, and generalization capabilities, are implicitly doing (unsupervised) big learning. Accordingly, we try to explain from the big learning perspective why they have such amazing characteristics.

- Firstly, by referring to Eq. (3) and Fig. 1, both the model parameters and latent features of $p_{\boldsymbol{\theta}}(\boldsymbol{x}_{\mathbb{T}}|\boldsymbol{x}_{\mathbb{S}})$ are shared among many/all data tasks,[5] manifested as a massive multi-task learning that exhaustively exploits the data information with statistical sharing power. Because all data tasks share a consistent goal to model (from diverse perspectives) the one underlying data distribution $q(\boldsymbol{x})$, it's expected that (unsupervised) big learning would encourage the parameters $\boldsymbol{\theta}$ (and also the latent features) to summarize the intrinsic data information or compositional data meta-knowledge (Wu et al., 2021; Lu et al., 2021), manifested as those amazing characteristics.

- Secondly, thanks to its comprehensive training nature, (unsupervised) big learning comes with extraordinary data and training-task flexibilities, enabling training with massive complete/incomplete data on many/all data tasks across potentially diverse domains. That significantly expanded training experiences (associated with both data and tasks) are expected to effectively reduce the training-test (or pretraining-finetuning) gap and therefore improves the robustness/generalization of big-learned foundation models.

**On the weighting of massive data tasks.** We highlight that (unsupervised) big learning comes with flexible weighting of its massive training tasks, *e.g.,* via predefined sampling strategy for $(\mathbb{S}, \mathbb{T})$. How to "optimally" weight those tasks is challenging and is likely downstream-task dependent; we will not cover it here expect presenting several thoughts. ($i$) Most real-world datasets naturally consist of both complete and incomplete samples, meaning the corresponding $(\mathbb{S}, \mathbb{T})$ pairs are already given; therefore, one may prefer to "*let the data speak for themselves.*" ($ii$) One can of course employ a specific sampling strategy for $(\mathbb{S}, \mathbb{T})$, *e.g.,* according to the available domain knowledge. It's worth emphasizing that, despite different weighting strategies, the optimum is the same, *i.e.,* $\boldsymbol{\theta}$ informationally identical to the underlying data distribution $q(\boldsymbol{x})$.

### 3.3 BIG LEARNING IN GENERAL SETTINGS

Based on unsupervised big learning with $\boldsymbol{X} = (\boldsymbol{x})$ containing only feature $\boldsymbol{x} \in \mathbb{R}^{L \times D}$, we next present its generalized version, *i.e.,* big learning, where $\boldsymbol{X} = (\boldsymbol{y}, \boldsymbol{x})$ contains both feature $\boldsymbol{x}$ and

---

[5]The diversity of the training objectives of such trustworthy data tasks is further enlarged in Section 3.3.

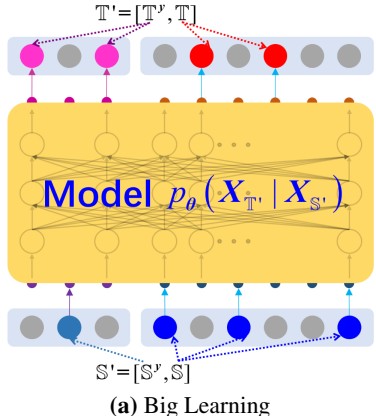
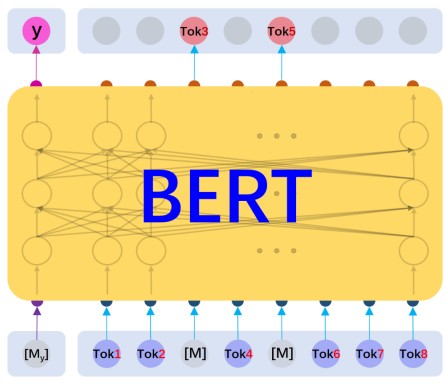

**(a)** Big Learning          **(b)** BERT (Stickland & Murray, 2019)

Figure 2: Big learning (a) and its special case of BERT (b). Similar to the mask token [M] for $\boldsymbol{x}$ (see Fig. 1b), we employ another mask token [M$_\text{y}$] for $\boldsymbol{y}$, which works identically to the classification token [CLS] in BERT settings (Stickland & Murray, 2019) and the start-of-sentence token in GPT settings (Brown et al., 2020). Often inserting [M]/[M$_\text{y}$] tokens later in a middle layer improves performance (He et al., 2021; Touvron et al., 2021).

supervision $\boldsymbol{y} \in \mathbb{R}^{L^{\boldsymbol{y}} \times D^{\boldsymbol{y}}}$. The *non-overlapping* source/target index subsets $\mathbb{S}/\mathbb{T}$ are correspondingly expanded as $\mathbb{S}' = [\mathbb{S}^{\boldsymbol{y}}, \mathbb{S}]$ and $\mathbb{T}' = [\mathbb{T}^{\boldsymbol{y}}, \mathbb{T}]$, respectively, where $\mathbb{L}' = [\mathbb{L}^{\boldsymbol{y}}, \mathbb{L}]$, $\mathbb{S}' \subset \mathbb{L}'$, $\mathbb{T}' \subseteq \mathbb{L}'$, and $\mathbb{T}' \neq \emptyset$.

Thanks to the modeling flexibility of unsupervised big learning, to generalize it into big learning is straight-forward, where the main idea is to model $p_{\boldsymbol{\theta}}(\boldsymbol{X}_{\mathbb{T}'}|\boldsymbol{X}_{\mathbb{S}'}) \longrightarrow q(\boldsymbol{X}_{\mathbb{T}'}|\boldsymbol{X}_{\mathbb{S}'})$ for many/all $(\mathbb{S}', \mathbb{T}')$ pairs, as demonstrated in Fig. 2a. $q(\boldsymbol{X}) \triangleq q(\boldsymbol{y}, \boldsymbol{x})$ is the underlying data distribution.

**Model architecture and training objective.** For situations where $\boldsymbol{X} = (\boldsymbol{y}, \boldsymbol{x})$ has the same data type (*e.g.,* both $\boldsymbol{y}$ and $\boldsymbol{x}$ denote a sequence of *continuous* features), big learning works basically the same as its unsupervised variant. We next elaborate on the situations with multimodality (Gupta et al., 2021; Li et al., 2021; Ramesh et al., 2021; 2022; Baevski et al., 2022), where *e.g.,* $\boldsymbol{y}$ denotes a *discrete* token sequence but $\boldsymbol{x}$ is a *continuous* feature sequence. We reveal two solutions.

1. **To transform one data type into the other for alignment**. For example, one can vector-quantize the *continuous* $\boldsymbol{x}$ into a sequence of *discrete* tokens, similar to the DALL-E (Ramesh et al., 2021), followed by employing similar techniques introduced in Section 3.1.

2. **To recursively reuse $p_{\boldsymbol{\theta}}(\boldsymbol{X}_{\mathbb{T}'}|\boldsymbol{X}_{\mathbb{S}'})$ to model the correlation between $\boldsymbol{x}$ and $\boldsymbol{y}$.** The key idea is to further exploit the flexibility of big learning. Specifically, we can unfold the learning via

$$p_{\boldsymbol{\theta}}(\boldsymbol{X}_{\mathbb{T}'}|\boldsymbol{X}_{\mathbb{S}'}) = p_{\boldsymbol{\theta}}(\boldsymbol{y}_{\mathbb{T}^{\boldsymbol{y}}}|\boldsymbol{x}_{\mathbb{T}}, \boldsymbol{X}_{\mathbb{S}'}) p_{\boldsymbol{\theta}}(\boldsymbol{x}_{\mathbb{T}}|\boldsymbol{X}_{\mathbb{S}'}) = p_{\boldsymbol{\theta}}(\boldsymbol{X}_{\mathbb{T}^{\boldsymbol{y}}}|\boldsymbol{X}_{\mathbb{T} \cup \mathbb{S}'}) p_{\boldsymbol{\theta}}(\boldsymbol{X}_{\mathbb{T}}|\boldsymbol{X}_{\mathbb{S}'}), \quad (6)$$

where $\boldsymbol{X}_{\mathbb{T}'} = (\boldsymbol{y}_{\mathbb{T}^{\boldsymbol{y}}}, \boldsymbol{x}_{\mathbb{T}})$ and $\boldsymbol{X}_{\mathbb{T}^{\boldsymbol{y}}}/\boldsymbol{X}_{\mathbb{T}}$ has one unique data type after unfolding. Big learning first forward-propagates twice through the model, with the output $\boldsymbol{x}_{\mathbb{T}}$ of the first propagation inserted to the input of the second one; after calculating the objective and thanks to the continuity of $\boldsymbol{x}_{\mathbb{T}}$, the gradients can be back-propagated to parameter $\boldsymbol{\theta}$ twice for model updating.

**BERT pretraining as a special case of big learning.** Section 3.1 revealed that (unsupervised) big learning contains the masked language modeling of the BERT pretraining as a special case. We next reveal that big learning also covers the next sentence prediction (NSP) task (Stickland & Murray, 2019), where $\boldsymbol{x}$ indicates two sentences and label $y \in \{0, 1\}$ indicates whether they are next to each other. It's readily verified that NSP is recovered with $\mathbb{S}' = [\emptyset, \mathbb{L}]$ and $\mathbb{T}' = [\{1\}, \emptyset]$ (refer to the 1st column of Table 2). To summarize, big learning contains the BERT pretraining as a special case.

**Big learning serves as a universal machine learning paradigm.** Benefiting from its modeling flexibility, big learning has most machine learning paradigms as special cases, as illustrated in Table 2. That universality of big learning, combined with its data/task flexibilities, enable flexible combinations and communications among different learning paradigms, *e.g.,* via the shared parameters $\boldsymbol{\theta}$ or training objectives like Eq. (5). Therefore, the proposed big learning might potentially facilitate semantically diverse multi-task self-learning on the Internet, producing brain-scale big/foundation models with reinforced performance, robustness, and generalization.

**Big learning versus self-supervised contrastive learning.** Contrastive learning focuses on exploiting domain prior knowledge to learn generally applicable data representations for downstream tasks (He

Table 2: Example special cases of big learning $p_{\boldsymbol{\theta}}(\boldsymbol{X}_{\mathbb{T}'}|\boldsymbol{X}_{\mathbb{S}'})$ with $\mathbb{S}' = [\mathbb{S}^y, \mathbb{S}]$ and $\mathbb{T}' = [\mathbb{T}^y, \mathbb{T}]$. Without loss of generality, we assume $y \in \{1, \cdots, C\}^{1 \times 1}$ is the label paired with $\boldsymbol{x}$, where $\mathbb{L}^y = \{1\}$. We focus on the core idea for demonstration and highlight that the model/objective implementations can be task-specific.

| Supervised Learning | Self-supervised Learning | Unconditioned Generation | Conditioned Generation |
|---|---|---|---|
| $p_{\boldsymbol{\theta}}(y|\boldsymbol{x})$ | $p_{\boldsymbol{\theta}}(\boldsymbol{x}_{\mathbb{T}}|\boldsymbol{x}_{\mathbb{S}})$ | $p_{\boldsymbol{\theta}}(\boldsymbol{x})$ | $p_{\boldsymbol{\theta}}(\boldsymbol{x}|y)$ |
| $\mathbb{S}' = [\emptyset, \mathbb{L}]$ | $\mathbb{S}' = [\emptyset, \mathbb{S}]$ | $\mathbb{S}' = [\emptyset, \emptyset]$ | $\mathbb{S}' = [\{1\}, \emptyset]$ |
| $\mathbb{T}' = [\{1\}, \emptyset]$ | $\mathbb{T}' = [\emptyset, \mathbb{T}]$ | $\mathbb{T}' = [\emptyset, \mathbb{L}]$ | $\mathbb{T}' = [\emptyset, \mathbb{L}]$ |

et al., 2020; Chen et al., 2020; Grill et al., 2020; Chen & He, 2021). From the perspective of prior exploitation, contrastive learning is orthogonal to our big learning that is mostly data-driven. One can of course consider leveraging the flexibility of big learning to combine it with contrastive learning for incorporating trustworthy domain priors; please refer to Appendix D for further discussions.

**On the *i.i.d.* assumption.** Due to the space constraint, please refer to Appendix E for the details.

# 4 EXPERIMENTS

The data/task flexibilities of big learning significantly expand its scope of application, which, however, also bring tremendous challenges to the comprehensive evaluations of its robustness, adaptability, and generalization capabilities. We emphasize that the great successes of existing big/foundation models have provided concrete evidences that support the presented big learning.

In what follows, we concentrate on demonstrating the main idea that (unsupervised) big learning is indeed capable of delivering many/all joint/conditional/marginal data capabilities, thanks to its extraordinary data/task flexibilities and exhaustive exploitation of data information. We conduct unsupervised big learning with all joint/conditional/marginal data tasks (via $(\mathbb{S}, \mathbb{T})$-sampling) on image datasets of MNIST and CelebA, based on Eqs. (4) and (5) (details are given in Appendix F). After training, we ($i$) diversely test the data completion capabilities of the big-learned model and ($ii$) ferociously challenge its generalization capability with abused anomalous out-of-domain tasks.

## 4.1 VERSATILE DATA COMPLETION CAPABILITIES WITH ADAPTIVE GENERATION DIVERSITY

We first test the big-learned data generation/completion capabilities with different ratios $r_{\mathbb{S}}$ of $\mathbb{S}$ in $\mathbb{L}$. For a specific $r_{\mathbb{S}}$, we either randomly sample $r_{\mathbb{S}}L$ image patches or choose the first $r_{\mathbb{S}}$-portion to form the source $\boldsymbol{x}_{\mathbb{S}}$, which is then input to the model $p_{\boldsymbol{\theta}}(\boldsymbol{x}_{\mathbb{T}}|\boldsymbol{x}_{\mathbb{S}})$ for image completion. Fig. 3 shows the corresponding results. It's clear that the big-learned model masters many/all joint/conditional/marginal data capabilities simultaneously. Besides, big learning also learns from the data an adaptive generation diversity conditioned on $\boldsymbol{x}_{\mathbb{S}}$. Specifically, with increasing/decreasing $r_{\mathbb{S}}$ (*i.e.,* more/less source information), big learning delivers increasingly deterministic/diverse generations controlled by $\boldsymbol{x}_{\mathbb{S}}$/random-noise, following our intuition (see Appendix H for more results).

We then test the big-learned capabilities with respect to various $\mathbb{S}$ and noise settings, with the results summarized in Fig. 4. On the one hand, given an image $\boldsymbol{x}$ and a random noise $\boldsymbol{z}$, big learning clearly delivers for various $\mathbb{S}$s diverse realistic generations on both MNIST (see the variations in class/stroke-

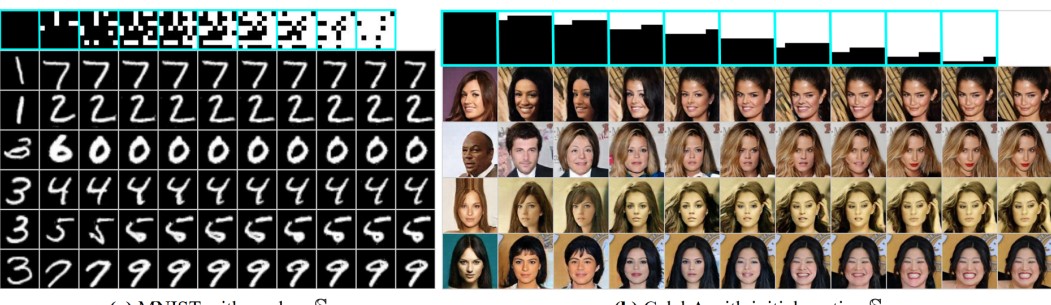

**(a)** MNIST with random $\mathbb{S}$        **(b)** CelebA with initial-portion $\mathbb{S}$

Figure 3: Versatile data generation/completion capabilities from big learning. The first row with light-blue boxes shows different $\mathbb{S}$s, with an increasing $\mathbb{S}$-ratio from left to right. The rightmost column gives the real image.

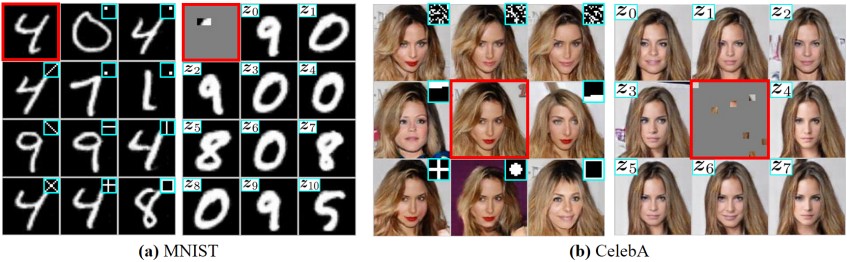

**(a)** MNIST      **(b)** CelebA

Figure 4: Versatile data completion capabilities from big learning *w.r.t.* various $\mathbb{S}$ (left) and noise $z$ (right). $\mathbb{S}$s are shown in upper-right light-blue boxes, while the red boxes show $x$ (left) and $x_{\mathbb{S}}$ (right), respectively.

thickness/shape/angle) and CelebA (see the varying identity/hair-style/make-up/expression). On the other hand, given a specific $x_{\mathbb{S}}$ with limited information, the big-learned model, when input different noises $z_i$, also generates realistic images with diversity.

The experimental results in Figs. 3 and 4 demonstrate that, by comprehensively exploiting the available information inherent in large-scale complete/incomplete data, big learning is capable of delivering versatile data generation/completion capabilities with learned adaptive generation diversity.

### 4.2 GENERALIZATION ON ABUSED ANOMALOUS OUT-OF-DOMAIN COMPLETION

We design abused completion tasks to ferociously challenge the generalization of our big learning. Specifically, we intentionally design $x_{\mathbb{S}}$ with ($i$) abused interventions to source patches (*e.g.,* random relocation and duplication, as shown in Fig. 5(a)); ($ii$) mixed-up patches from different data samples (see Fig. 5(b)); and ($iii$) unseen out-of-domain image patches, as shown in Figs. 5(c)-(d).

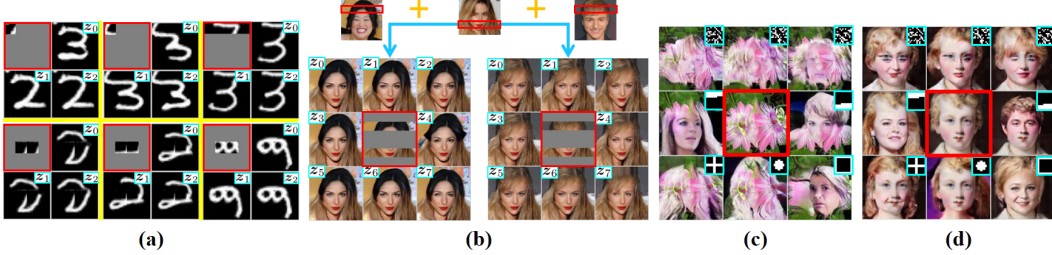

**(a)**      **(b)**      **(c)**      **(d)**

Figure 5: Abused anomalous completion for demonstrating the generalization of big learning. (a) $x_{\mathbb{S}}$ constructed with random center patches replaced in the upper-left corner (top) and duplicated and replaced in the center (bottom). A model big-learned on CelebA is used in (b)-(d). (b) $x_{\mathbb{S}}$ combining patches from different images. Out-of-domain $x_{\mathbb{S}}$ from Flowers (Nilsback & Zisserman, 2008) (c) and MetFaces (Karras et al., 2020) (d).

It's clear that big learning manages to handle these abused $x_{\mathbb{S}}$ with reasonable image completion; *e.g.,* see the realistic characters with overall consistent style and smooth strokes in Fig. 5(a), the harmoniously completed faces even with mismatched face frame and hair color in Fig. 5(b), and the fluent out-of-domain completion with smooth junctions in Figs. 5(c)-(d). These surprising results from abused anomalous out-of-domain completions (as well as the great successes of existing foundation models) justify the remarkable generalization capability of the presented big learning.

## 5 CONCLUSIONS

We propose the big learning that exhaustively exploits the available data information during training and potentially delivers all joint/conditional/marginal data capabilities after training. We reveal that big learning ($i$) comes with marvelous training flexibilities for complete/incomplete data and for customizing training tasks, ($ii$) is what existing foundation models are implicitly doing, and ($iii$) unifies conventional machine learning paradigms and enables their flexible cooperations. Though inspiring, big learning also shares the constrains of foundation models that are discussed in detail in Bommasani et al. (2021); Yuan et al. (2022); *e.g.,* to comprehensively verify its effectiveness (for general downstream tasks) is extremely challenging and time-consuming. Therefore, we believe that big learning needs our community, and vice versa.

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

## Appendix of
## Big Learning: A Universal Machine Learning Paradigm?

**Anonymous Authors**

## A  ON NAIVE MODELING OF ALL JOINT/CONDITIONAL/MARGINAL DATA DISTRIBUTIONS

We present with the unsupervised settings, where $\boldsymbol{x} \in \mathbb{R}^{L \times D}$ with length $L$ and dimension $D$ (like $L$ flattened patches of an image or $L$ words with $D = 1$). It's straightforward to generalize the following analyses to the general settings with a data sample $\boldsymbol{X} = (\boldsymbol{y}, \boldsymbol{x})$ contains an additional supervision $\boldsymbol{y} \in \mathbb{R}^{L^{\boldsymbol{y}} \times D^{\boldsymbol{y}}}$. Considering $D > 1$ and $D = 1$ for image patches and text words, respectively, we concentrate on analyzing the modeling of all joint/conditional/marginal data distributions *w.r.t.* the length $L$ below.

As mentioned in the main manuscript, one need construct $N_{\text{all}} = \sum_{i=0}^{L-1} C_L^i (\sum_{k=1}^{L-i} C_{L-i}^k)$ models to naively model all joint/conditional/marginal data distributions, to collect all joint/conditional/marginal data capabilities. $C_L^i$ denotes the number of $i$-combinations from a set with $L$ elements.

To elaborate on that, consider a simple 3-length 1-dimensional problem with $\boldsymbol{x} = [x_1, x_2, x_3]^T$, where $L = 3$, $D = 1$, $x_i \in \mathbb{R}$, and the length index set $\mathbb{L} = \{1, 2, 3\}$.

- The goal of the joint modeling is to deliver $p_{\boldsymbol{\theta}}(\boldsymbol{x}) \longrightarrow q(\boldsymbol{x})$ with one model $p_{\boldsymbol{\theta}}(\boldsymbol{x})$.
- By contrast, to naively model all joint/conditional/marginal data distributions, one need construct 19 models for such a simple 3-length problem, *i.e.,*

$$
\begin{aligned}
&p_{\boldsymbol{\theta}^1}(x_1),\, p_{\boldsymbol{\theta}^2}(x_2),\, p_{\boldsymbol{\theta}^3}(x_3),\, p_{\boldsymbol{\theta}^4}(x_1, x_2),\, p_{\boldsymbol{\theta}^5}(x_2, x_3),\, p_{\boldsymbol{\theta}^6}(x_1, x_3),\, p_{\boldsymbol{\theta}^7}(x_1, x_2, x_3), \\
&p_{\boldsymbol{\theta}^8}(x_2|x_1),\, p_{\boldsymbol{\theta}^9}(x_3|x_1),\, p_{\boldsymbol{\theta}^{10}}(x_2, x_3|x_1), \\
&p_{\boldsymbol{\theta}^{11}}(x_1|x_2),\, p_{\boldsymbol{\theta}^{12}}(x_3|x_2),\, p_{\boldsymbol{\theta}^{13}}(x_1, x_3|x_2), \\
&p_{\boldsymbol{\theta}^{14}}(x_1|x_3),\, p_{\boldsymbol{\theta}^{15}}(x_2|x_3),\, p_{\boldsymbol{\theta}^{16}}(x_1, x_2|x_3), \\
&p_{\boldsymbol{\theta}^{17}}(x_1|x_2, x_3),\, p_{\boldsymbol{\theta}^{18}}(x_2|x_1, x_3),\, p_{\boldsymbol{\theta}^{19}}(x_3|x_1, x_2).
\end{aligned}
\tag{7}
$$

Based on the above 3-length problem, one can readily summarize the following two steps in calculating the number of models in naively modeling all joint/conditional/marginal data distributions, *i.e.,* $q(\boldsymbol{x}_{\mathbb{T}}|\boldsymbol{x}_{\mathbb{S}}), \forall \mathbb{S} \subset \mathbb{L}, \mathbb{T} \subseteq \mathbb{L}, \mathbb{T} \neq \emptyset$.

1. **Sample $\mathbb{S}$.** The source index set $\mathbb{S}$ may contain $\{0, \cdots, L-1\}$ indexes/locations, where $\mathbb{S}$ containing 0 index corresponds to joint/marginal generations and $\mathbb{S}$ containing $\geq 1$ indexes corresponds to conditional generations/completions. For a special case with $i$ indexes in $\mathbb{S}$ with $i \in [0, L-1]$, one has $C_L^i$ ways to specify that source index set $\mathbb{S}$.

2. **Sample $\mathbb{T}$ conditioned on $\mathbb{S}$.** Given a $\mathbb{S}$ consisting of $i$ indexes, the target index set $\mathbb{T}$ could contain $\{1, \cdots, L-i\}$ indexes/locations outside $\mathbb{S}$. For a special case of $\mathbb{T}$ containing $k$ indexes where $k \in [1, L-i]$, one has $C_{L-i}^k$ ways to specify the target $\mathbb{T}$.

Therefore, to naively model all joint/conditional/marginal data distributions, one need construct $N_{\text{all}} = \sum_{i=0}^{L-1} C_L^i (\sum_{k=1}^{L-i} C_{L-i}^k)$ models, which, however, is prohibitive in practice.

Note with ideal modeling of $q(\boldsymbol{x}_{\mathbb{T}}|\boldsymbol{x}_{\mathbb{S}})$, the orders in $\mathbb{S}/\mathbb{T}$ should not matter. However, that may not hold true considering practical constraints, *e.g.,* where existing joint modeling techniques fail to model the multi-mode characteristics of $\boldsymbol{x}_{\mathbb{T}}$. Besides, in the NLP application of language modeling, one may be interested in versatile (conditional) generation ordering (as defined in $\mathbb{T}$), mimicking the permutation language modeling (Yang et al., 2019). In that case, to naively modeling all joint/conditional/marginal data distributions, one need construct $N'_{\text{all}} = \sum_{i=0}^{L-1} C_L^i (\sum_{k=1}^{L-i} A_{L-i}^k)$ models to take into consideration the order of $\mathbb{T}$, where the order of $\mathbb{S}$ is ignored and $A_{L-i}^k$ denotes the number of the ordered arrangements of $k$ elements from a set with $L-1$ elements. Similarly, one need construct $N''_{\text{all}} = \sum_{i=0}^{L-1} A_L^i (\sum_{k=1}^{L-i} A_{L-i}^k)$ models to model the orders in both $\mathbb{S}$ and $\mathbb{T}$.

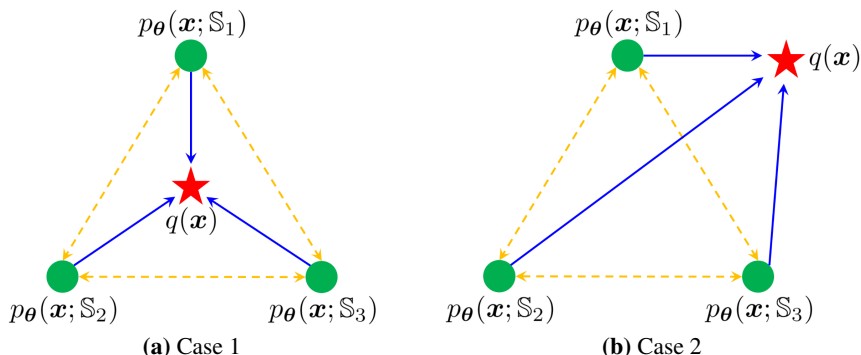

**(a)** Case 1            **(b)** Case 2

Figure 6: Demonstration of unsupervised big learning based on GANs.

# B   DERIVATIONS OF THE GAN EXAMPLE ASSOCIATED WITH EQS. (4) AND (5)

Here we present the detailed derivations/proofs for the GAN example associated with Eqs. (4) and (5) of the main manuscript. For better understanding, we begin with a simplified case where $\mathbb{T} = \mathbb{L} \backslash \mathbb{S}$, followed by generalizing the results to the general situations with $\mathbb{T} \subseteq \mathbb{L} \backslash \mathbb{S}$.

## B.1   $\mathbb{T} = \mathbb{L} \backslash \mathbb{S}$

To leverage the GAN training framework (Goodfellow et al., 2014), one needs the sampling capabilities from the distributions of interest. With $\mathbb{T} = \mathbb{L} \backslash \mathbb{S}$, here we are interested in the joint distributions with accessible sampling capabilities, including

$$
\begin{aligned}
&q(\boldsymbol{x}) \\
&p_{\boldsymbol{\theta}}(\boldsymbol{x}; \mathbb{S}) = p_{\boldsymbol{\theta}}(\boldsymbol{x}_{\mathbb{L}\backslash\mathbb{S}} | \boldsymbol{x}_{\mathbb{S}}) q(\boldsymbol{x}_{\mathbb{S}}) \quad \forall \mathbb{S}.
\end{aligned}
\tag{8}
$$

Note one can of course exploit the flexibility of big learning to define other joint distributions with sampling capabilities, such as an recursively defined distribution

$$
p_{\boldsymbol{\theta}}(\boldsymbol{x}; \mathbb{S}^1, \mathbb{S}^2) = p_{\boldsymbol{\theta}}(\boldsymbol{x}_{\mathbb{L}\backslash\mathbb{S}^2} | \boldsymbol{x}_{\mathbb{S}^2}) p_{\boldsymbol{\theta}}(\boldsymbol{x}_{\mathbb{S}^2}),
\tag{9}
$$

where $p_{\boldsymbol{\theta}}(\boldsymbol{x}_{\mathbb{S}^2}) = \int p_{\boldsymbol{\theta}}(\boldsymbol{x}_{\mathbb{L}\backslash\mathbb{S}^1} | \boldsymbol{x}_{\mathbb{S}^1}) q(\boldsymbol{x}_{\mathbb{S}^1}) d\boldsymbol{x}_{\mathbb{L}\backslash\mathbb{S}^2}$. For simplicity, we focus on the simplified settings in Eq. (8) and leave the interesting but complicated recursive case for future research.

Given the underlying data distribution $q(\boldsymbol{x})$ and "model" distributions $p_{\boldsymbol{\theta}}(\boldsymbol{x}; \mathbb{S})$ in Eq. (8),

1. one can match any $p_{\boldsymbol{\theta}}(\boldsymbol{x}; \mathbb{S})$ to $q(\boldsymbol{x})$ adversarially with a GAN. Take the standard GAN (Goodfellow et al., 2014) for an example, the objective is

$$
\min_{\boldsymbol{\theta}} \max_{\boldsymbol{\phi}} \mathbb{E}_{q(\boldsymbol{x})} \log \sigma(f_{\boldsymbol{\phi}}(\boldsymbol{x}; \mathbb{S})) + \mathbb{E}_{p_{\boldsymbol{\theta}}(\boldsymbol{x}_{\mathbb{L}\backslash\mathbb{S}} | \boldsymbol{x}_{\mathbb{S}}) q(\boldsymbol{x}_{\mathbb{S}})} \log(1 - \sigma(f_{\boldsymbol{\phi}}(\boldsymbol{x}; \mathbb{S}))),
\tag{10}
$$

where the optimal $f_{\boldsymbol{\phi}^*}(\boldsymbol{x}; \mathbb{S}) = \log \frac{q(\boldsymbol{x})}{p_{\boldsymbol{\theta}}(\boldsymbol{x}_{\mathbb{L}\backslash\mathbb{S}} | \boldsymbol{x}_{\mathbb{S}}) q(\boldsymbol{x}_{\mathbb{S}})} = \log \frac{q(\boldsymbol{x}_{\mathbb{L}\backslash\mathbb{S}} | \boldsymbol{x}_{\mathbb{S}})}{p_{\boldsymbol{\theta}}(\boldsymbol{x}_{\mathbb{L}\backslash\mathbb{S}} | \boldsymbol{x}_{\mathbb{S}})}$. Ideally, optimizing the above objective is identical to minimizing the Jensen-Shannon divergence $\mathrm{JS}[q(\boldsymbol{x}) || p_{\boldsymbol{\theta}}(\boldsymbol{x}; \mathbb{S})]$, as illustrated with the blue solid arrows in Fig. 6.

2. one can also conduct matching among any two model distributions (*e.g.*, $p_{\boldsymbol{\theta}}(\boldsymbol{x}; \mathbb{S}^1) = p_{\boldsymbol{\theta}}(\boldsymbol{x}_{\mathbb{L}\backslash\mathbb{S}^1} | \boldsymbol{x}_{\mathbb{S}^1}) q(\boldsymbol{x}_{\mathbb{S}^1})$ and $p_{\boldsymbol{\theta}}(\boldsymbol{x}; \mathbb{S}^2) = p_{\boldsymbol{\theta}}(\boldsymbol{x}_{\mathbb{L}\backslash\mathbb{S}^2} | \boldsymbol{x}_{\mathbb{S}^2}) q(\boldsymbol{x}_{\mathbb{S}^2})$) to enable communications/cooperations among them, via optimizing

$$
\min_{\boldsymbol{\theta}} \max_{\boldsymbol{\phi}} \begin{cases} \mathbb{E}_{p_{\boldsymbol{\theta}}(\boldsymbol{x}_{\mathbb{L}\backslash\mathbb{S}^1} | \boldsymbol{x}_{\mathbb{S}^1}) q(\boldsymbol{x}_{\mathbb{S}^1})} \log \sigma(f'_{\boldsymbol{\phi}}(\boldsymbol{x}; \mathbb{S}^1, \mathbb{S}^2)) \\ + \mathbb{E}_{p_{\boldsymbol{\theta}}(\boldsymbol{x}_{\mathbb{L}\backslash\mathbb{S}^2} | \boldsymbol{x}_{\mathbb{S}^2}) q(\boldsymbol{x}_{\mathbb{S}^2})} \log(1 - \sigma(f'_{\boldsymbol{\phi}}(\boldsymbol{x}; \mathbb{S}^1, \mathbb{S}^2))) \end{cases}
\tag{11}
$$

where the optimal $f'_{\boldsymbol{\phi}^*}(\boldsymbol{x}; \mathbb{S}^1, \mathbb{S}^2) = \log \frac{p_{\boldsymbol{\theta}}(\boldsymbol{x}_{\mathbb{L}\backslash\mathbb{S}^1} | \boldsymbol{x}_{\mathbb{S}^1}) q(\boldsymbol{x}_{\mathbb{S}^1})}{p_{\boldsymbol{\theta}}(\boldsymbol{x}_{\mathbb{L}\backslash\mathbb{S}^2} | \boldsymbol{x}_{\mathbb{S}^2}) q(\boldsymbol{x}_{\mathbb{S}^2})}$. The orange dotted arrows in Fig. 6 demonstrate such idea.

At first sight of Eqs. (10) and (11), it seems one should at least construct two discriminators, with $f_\phi(\boldsymbol{x};\mathbb{S})$ and $f'_\phi(\boldsymbol{x};\mathbb{S}^1,\mathbb{S}^2)$ respectively. However, we notice that

$$
\begin{aligned}
f'_{\phi^*}(\boldsymbol{x};\mathbb{S}^1,\mathbb{S}^2) &= \log\frac{q(\boldsymbol{x})}{p_{\boldsymbol{\theta}}(\boldsymbol{x}_{\mathbb{L}\setminus\mathbb{S}^2}|\boldsymbol{x}_{\mathbb{S}^2})q(\boldsymbol{x}_{\mathbb{S}^2})} - \log\frac{q(\boldsymbol{x})}{p_{\boldsymbol{\theta}}(\boldsymbol{x}_{\mathbb{L}\setminus\mathbb{S}^1}|\boldsymbol{x}_{\mathbb{S}^1})q(\boldsymbol{x}_{\mathbb{S}^1})} \\
&= f_{\phi^*}(\boldsymbol{x};\mathbb{S}^2) - f_{\phi^*}(\boldsymbol{x};\mathbb{S}^1).
\end{aligned}
$$

Accordingly, we propose to employ further simplification that builds $f'_\phi(\boldsymbol{x};\mathbb{S}^1,\mathbb{S}^2)$ on top of $f_\phi(\boldsymbol{x};\mathbb{S})$, *i.e.,* we reformulate Eq. (11) as

$$
\min_{\boldsymbol{\theta}}\max_{\phi}\begin{cases}\mathbb{E}_{p_{\boldsymbol{\theta}}(\boldsymbol{x}_{\mathbb{L}\setminus\mathbb{S}^1}|\boldsymbol{x}_{\mathbb{S}^1})q(\boldsymbol{x}_{\mathbb{S}^1})}\log\sigma[f_\phi(\boldsymbol{x};\mathbb{S}^2) - f_\phi(\boldsymbol{x};\mathbb{S}^1)] \\ + \mathbb{E}_{p_{\boldsymbol{\theta}}(\boldsymbol{x}_{\mathbb{L}\setminus\mathbb{S}^2}|\boldsymbol{x}_{\mathbb{S}^2})q(\boldsymbol{x}_{\mathbb{S}^2})}\log\sigma[f_\phi(\boldsymbol{x};\mathbb{S}^1) - f_\phi(\boldsymbol{x};\mathbb{S}^2)].\end{cases} \tag{12}
$$

Till now, we present the derivations associated with $\mathbb{T}=\mathbb{L}\setminus\mathbb{S}$, *i.e.,* matching in the joint space. In what follows, we generalize to the settings with $\mathbb{T}\subseteq\mathbb{L}\setminus\mathbb{S}$, to deliver (unsupervised) big learning in all joint/conditional/marginal spaces.

## B.2 $\mathbb{T}\subseteq\mathbb{L}\setminus\mathbb{S}$

Similar to the previous section, we also consider simplified situations with no recursiveness, that is, we do not consider a model distribution $p_{\boldsymbol{\theta}}(\boldsymbol{x}_{\mathbb{T}}|\boldsymbol{x}_{\mathbb{S}})p_{\boldsymbol{\theta}}(\boldsymbol{x}_{\mathbb{S}})$, even though such recursive flexibility of big learning is quite interesting. We leave that as future research.

Accordingly, the considered joint/conditional/marginal distributions with sampling capabilities are

$$
\begin{aligned}
q(\boldsymbol{x}_{\mathbb{S}\cup\mathbb{T}}) & \\
p_{\boldsymbol{\theta}}(\boldsymbol{x}_{\mathbb{S}\cup\mathbb{T}}) = p_{\boldsymbol{\theta}}(\boldsymbol{x}_{\mathbb{T}}|\boldsymbol{x}_{\mathbb{S}})q(\boldsymbol{x}_{\mathbb{S}}) &\qquad \forall\mathbb{S},\mathbb{T}
\end{aligned} \tag{13}
$$

where $\mathbb{S}\cup\mathbb{T}$ need not be $\mathbb{L}$. Note $\mathbb{S}\cup\mathbb{T}\subset\mathbb{L}$ means the corresponding $q(\boldsymbol{x}_{\mathbb{S}\cup\mathbb{T}})$ is a *marginal* data distribution, whose data samples are readily accessible from those of $q(\boldsymbol{x})$.

Similar to the previous section,

- one can match any model distribution $p_{\boldsymbol{\theta}}(\boldsymbol{x}_{\mathbb{S}\cup\mathbb{T}})$ to the underlying joint/marginal data distribution $q(\boldsymbol{x}_{\mathbb{S}\cup\mathbb{T}})$, via the standard GAN objective

$$
\min_{\boldsymbol{\theta}}\max_{\phi}\mathbb{E}_{q(\boldsymbol{x}_{\mathbb{S}\cup\mathbb{T}})}\log\sigma(f_\phi(\boldsymbol{x};\mathbb{S},\mathbb{T})) + \mathbb{E}_{p_{\boldsymbol{\theta}}(\boldsymbol{x}_{\mathbb{T}}|\boldsymbol{x}_{\mathbb{S}})q(\boldsymbol{x}_{\mathbb{S}})}\log(1 - \sigma(f_\phi(\boldsymbol{x};\mathbb{S},\mathbb{T}))), \tag{14}
$$

  where $f_{\phi^*}(\boldsymbol{x};\mathbb{S},\mathbb{T}) = \log\frac{q(\boldsymbol{x}_{\mathbb{S}\cup\mathbb{T}})}{p_{\boldsymbol{\theta}}(\boldsymbol{x}_{\mathbb{T}}|\boldsymbol{x}_{\mathbb{S}})q(\boldsymbol{x}_{\mathbb{S}})} = \log\frac{q(\boldsymbol{x}_{\mathbb{T}}|\boldsymbol{x}_{\mathbb{S}})}{p_{\boldsymbol{\theta}}(\boldsymbol{x}_{\mathbb{T}}|\boldsymbol{x}_{\mathbb{S}})}$.

- one can also conduct matching among any two model distributions, *e.g.,* $p_{\boldsymbol{\theta}}(\boldsymbol{x}_{\mathbb{T}^1}|\boldsymbol{x}_{\mathbb{S}^1})q(\boldsymbol{x}_{\mathbb{S}^1})$ and $p_{\boldsymbol{\theta}}(\boldsymbol{x}_{\mathbb{T}^2}|\boldsymbol{x}_{\mathbb{S}^2})q(\boldsymbol{x}_{\mathbb{S}^2})$, as long as $\mathbb{S}^1\cup\mathbb{T}^1 = \mathbb{S}^2\cup\mathbb{T}^2$, with the corresponding objective

$$
\min_{\boldsymbol{\theta}}\max_{\phi}\begin{cases}\mathbb{E}_{p_{\boldsymbol{\theta}}(\boldsymbol{x}_{\mathbb{T}^1}|\boldsymbol{x}_{\mathbb{S}^1})q(\boldsymbol{x}_{\mathbb{S}^1})}\log\sigma(f_\phi(\boldsymbol{x};\mathbb{S}^1,\mathbb{T}^1,\mathbb{S}^2,\mathbb{T}^2)) \\ + \mathbb{E}_{p_{\boldsymbol{\theta}}(\boldsymbol{x}_{\mathbb{T}^2}|\boldsymbol{x}_{\mathbb{S}^2})q(\boldsymbol{x}_{\mathbb{S}^2})}\log(1 - \sigma(f_\phi(\boldsymbol{x};\mathbb{S}^1,\mathbb{T}^1,\mathbb{S}^2,\mathbb{T}^2))),\end{cases} \tag{15}
$$

  where $f'_{\phi^*}(\boldsymbol{x};\mathbb{S}^1,\mathbb{T}^1,\mathbb{S}^2,\mathbb{T}^2) = \log\frac{p_{\boldsymbol{\theta}}(\boldsymbol{x}_{\mathbb{T}^1}|\boldsymbol{x}_{\mathbb{S}^1})q(\boldsymbol{x}_{\mathbb{S}^1})}{p_{\boldsymbol{\theta}}(\boldsymbol{x}_{\mathbb{T}^2}|\boldsymbol{x}_{\mathbb{S}^2})q(\boldsymbol{x}_{\mathbb{S}^2})}$.

  For further simplifications, we again resort to

$$
\begin{aligned}
f'_{\phi^*}(\boldsymbol{x};\mathbb{S}^1,\mathbb{T}^1,\mathbb{S}^2,\mathbb{T}^2) &= \log\frac{q(\boldsymbol{x}_{\mathbb{S}^2\cup\mathbb{T}^2})}{p_{\boldsymbol{\theta}}(\boldsymbol{x}_{\mathbb{T}^2}|\boldsymbol{x}_{\mathbb{S}^2})q(\boldsymbol{x}_{\mathbb{S}^2})} - \log\frac{q(\boldsymbol{x}_{\mathbb{S}^1\cup\mathbb{T}^1})}{p_{\boldsymbol{\theta}}(\boldsymbol{x}_{\mathbb{T}^1}|\boldsymbol{x}_{\mathbb{S}^1})q(\boldsymbol{x}_{\mathbb{S}^1})} \\
&= f_{\phi^*}(\boldsymbol{x};\mathbb{S}^2,\mathbb{T}^2) - f_{\phi^*}(\boldsymbol{x};\mathbb{S}^1,\mathbb{T}^1)
\end{aligned}
$$

  and build $f'_\phi(\boldsymbol{x};\mathbb{S}^1,\mathbb{T}^1,\mathbb{S}^2,\mathbb{T}^2)$ on top of $f_\phi(\boldsymbol{x};\mathbb{S},\mathbb{T})$.

  Accordingly, Eq. (15) is reformulated as

$$
\min_{\boldsymbol{\theta}}\max_{\phi}\begin{cases}\mathbb{E}_{p_{\boldsymbol{\theta}}(\boldsymbol{x}_{\mathbb{T}^1}|\boldsymbol{x}_{\mathbb{S}^1})q(\boldsymbol{x}_{\mathbb{S}^1})}\log\sigma[f_\phi(\boldsymbol{x};\mathbb{S}^2,\mathbb{T}^2) - f_\phi(\boldsymbol{x};\mathbb{S}^1,\mathbb{T}^1)] \\ + \mathbb{E}_{p_{\boldsymbol{\theta}}(\boldsymbol{x}_{\mathbb{T}^2}|\boldsymbol{x}_{\mathbb{S}^2})q(\boldsymbol{x}_{\mathbb{S}^2})}\log\sigma[f_\phi(\boldsymbol{x};\mathbb{S}^1,\mathbb{T}^1) - f_\phi(\boldsymbol{x};\mathbb{S}^2,\mathbb{T}^2)].\end{cases} \tag{16}
$$

Accordingly, we conclude the proofs for the GAN example of the main manuscript.

# C    ON MODEL ARCHITECTURES OF THE GAN EXAMPLE IN EQS. (4) AND (5)

We next focus on discussing the model architectures of the GAN generator and discriminator employed in Eqs. (14) and (16) (*i.e.,* Eqs. (4) and (5) of the main manuscript).

Recently, the community begins to exploit integrating ViTs into GANs (Jiang et al., 2021; Lee et al., 2021; Zhao et al., 2021; Zhang et al., 2021). For example, the ViTGAN (Lee et al., 2021), delivering SOTA generative performance, employs simple modifications to the ViT architecture to construct the generator and the discriminator, but adopts *many* techniques to regularize the ViT-based discriminator for stable training. Motivated by the modeling flexibility of ViTs, we also employ ViT-based GAN generator and discriminator in the experiments, but similarly, find it challenging to stabilize GAN training with a ViT-based discriminator. It's worth highlighting that it's possible to design other alternative model architectures for our big learning; we employ what's presented below for a demonstration.

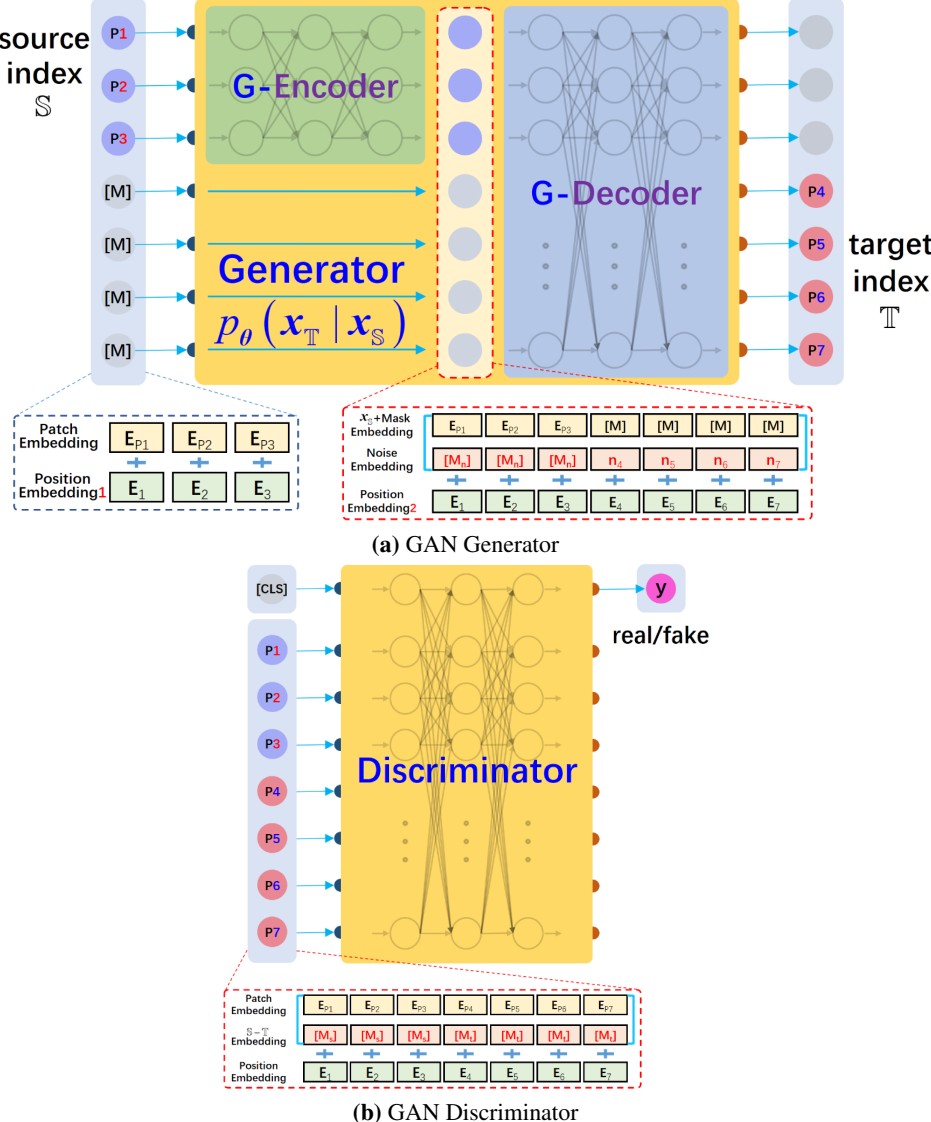

**(a)** GAN Generator

**(b)** GAN Discriminator

Figure 7: Example implementations of the GAN generator and discriminator employed in Eqs. (14) and (16) (*i.e.,* Eqs. (4) and (5) of the main manuscript).

Fig. 7 demonstrates the employed GAN generator and discriminator, both of which are constructed with Transformers/ViTs to exploit their modeling capabilities and flexibilities.

- **GAN Generator.** Following the MAE (He et al., 2021), we design the GAN generator $p_{\boldsymbol{\theta}}(\boldsymbol{x}_{\mathbb{T}}|\boldsymbol{x}_{\mathbb{S}})$ with an autoencoder-like architecture, which employs an encoding G-Encoder and a decoding G-Decoder, as shown in Fig. 7a. The G-Encoder encodes the source patches $\boldsymbol{x}_{\mathbb{S}}$ (if any) to their latent codes; then, these codes are combined with the mask tokens [M], patch-wise noise embeddings, and new positional encodings to serve as the input of the G-Decoder; finally, the G-Decoder transforms its input to generate the target patches $\boldsymbol{x}_{\mathbb{T}}$.

  [M] tokens are inserted later in a middle layer, because doing this often improves performance and lowers the computational burden (Touvron et al., 2021; He et al., 2021). A noise $\boldsymbol{z}$ is mapped with an 8-layer MLP to produce the patch-wise noise embeddings $\{\boldsymbol{n}_1, \cdots, \boldsymbol{n}_L\}$. Note we also introduce another toke [M$_n$] to indicate no noise embeddings are necessary at the corresponding source locations in $\mathbb{S}$.

- **GAN Discriminator.** As shown in Fig. 7b, we also modify the Transformer/ViT architecture to construct the universal GAN discriminator $\sigma(f_{\boldsymbol{\phi}}(\boldsymbol{x}; \mathbb{S}, \mathbb{T}))$ that applies to all $(\mathbb{S}, \mathbb{T})$ cases. We employ an additional CLS token mimicking the BERT, whose output indicates whether the input patches are realistic or not (more specifically, whether they form a "real" data from $q(\boldsymbol{x}_{\mathbb{S} \cup \mathbb{T}})$ or a fake one from $p_{\boldsymbol{\theta}}(\boldsymbol{x}_{\mathbb{T}}|\boldsymbol{x}_{\mathbb{S}})q(\boldsymbol{x}_{\mathbb{S}})$, by referring to Eq. (14)). The input of the discriminator consists of patch embeddings, positional embeddings, and two new special tokens ([M$_s$] and [M$_t$]) that indicate source or target patches mimicking the sentence tokens in the BERT.

## D  BIG LEARNING VERSUS CONTRASTIVE LEARNING

Contrastive learning (Hadsell et al., 2006) aims at learning a latent representation space, where the representations of different views of the same image ("positive pairs") are near each other but those from different images ("negative pairs") are far away from each other.

As discussed in the main manuscript, the self-supervised contrastive learning focuses on exploiting domain prior knowledge to learn generally applicable data representations, while the presented big learning is mostly data-driven. From that perspective, they are orthogonal to each other. However, we reveal below that big learning and contrastive learning have a lot in common.

- Both of them are based on massive multi-task training, associated with source/target indexes $(\mathbb{S}, \mathbb{T})$ and online/target augmentation pairs $(\mathcal{A}, \mathcal{B})$ (see Fig. 8a), respectively.
- Both of them share a universal model among massive training tasks.
- From the information perspective, both of them predict (or retrieve) the $\mathbb{T}/\mathcal{B}$-associated information conditioned on the information related to $\mathbb{S}/\mathcal{A}$, as detailed below.

Existing contrastive learning methods can be roughly grouped into two groups, based on whether the method uses negative pairs (like SimCLR (Chen et al., 2020) and MoCo (Chen et al., 2021b)) or not (like BYOL (Grill et al., 2020) and SimSiam (Chen & He, 2021)).

- **Group 1.** Contrastive learning methods using negative pairs, like SimCLR and MoCo, can be interpreted as *retrieving* (in the latent representation space) the target positively paired sample $\boldsymbol{B}_i$ from the negative samples within the mini-batch, conditioned on the source augmented sample $\boldsymbol{A}_i$, as illustrated in Fig. 8a.
- **Group 2.** Contrastive learning methods not using negative pairs, like BYOL and SimSiam, directly predict/generate (in the latent representation space) the target/teacher projection associated with $\boldsymbol{B}_i$, conditioned on the student projection associated with $\boldsymbol{A}_i$, as demonstrated in Fig. 8b.

Either group of contrastive learning methods retrieves or predicts the $\mathcal{B}$-associated information conditioned on the information related to $\mathcal{A}$, which is quite similar to the proposed big learning that predicts/generates $\boldsymbol{x}_{\mathbb{T}}$ conditioned on $\boldsymbol{x}_{\mathbb{S}}$. Therefore, from the information perspective, both of them predict (or retrieve) a piece of data/prior information conditioned on another piece of data/prior information.

It's interesting to consider combining big learning with contrastive learning to exhaustively exploit the available information from both data and domain-prior perspectives.

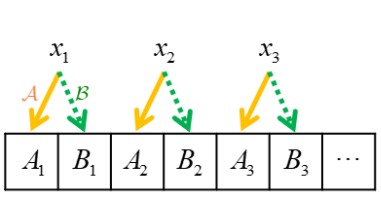
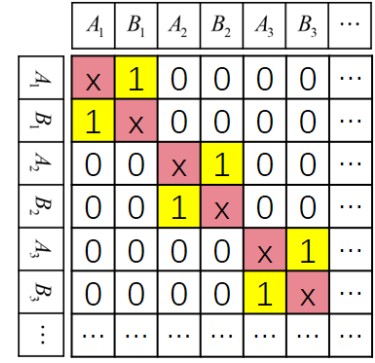

**(a)** Group 1 predicts/retrieves the positively paired sample from the negative samples within a mini-batch. $\mathcal{A}$ and $\mathcal{B}$ denote the online and target augmentation, respectively.

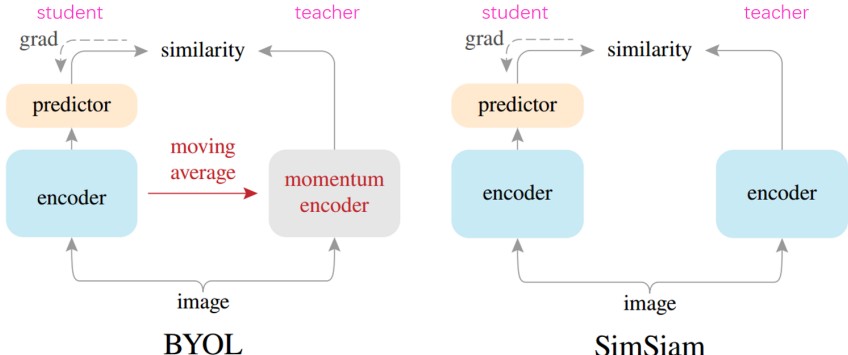

**(b)** Group 2 directly generates/predicts the target/teacher projection conditioned on the student projection.

Figure 8: Demonstrations of contrastive learning methods. (b) is adapted from Fig. 3 of Chen & He (2021).

## E  ON THE *i.i.d.* ASSUMPTION

The *i.i.d.* assumption is one of the key foundations of deep learning. But it's also well-known that the training data collected for practical applications are rarely *i.i.d.*, leading to a training-test gap (*i.e.,* novel/unseen data information emerges during testing) that significantly hinders the practical reliability of existing deep-learning models.

Recently, foundation models begin to demonstrate increasing robustness and generalization towards that gap, likely due to their large-scale pretraining (special cases of big learning) effectively reduces the probability of observing novel information during testing. We next elaborate on that and present 3 reasons on why big learning is expected to significantly reduce the training-test/pretraining-finetuning gap for improved robustness.

1. Thanks to its remarkable flexibilities on complete/incomplete training data and massive diverse learning tasks, big learning delivers abundant "exercise experiences" that significantly enlarges the training scope of the model, making it less likely to be "surprised" by novel test data/tasks.

2. Manually collecting or filtering data samples will likely introduce unintentional interventions that violate the *i.i.d.* assumption. The data flexibility of big learning makes it possible to conduct training with minimal human interventions in data collection, and accordingly, "let the data speak for themselves."

3. Even with the same dataset with all complete samples, big learning is expected to behave more robustly to the *i.i.d.* assumption, because ($i$) to collect perfectly *i.i.d.* complete samples is often intractable for practical applications; ($ii$) the conditional tasks of big learning are always implemented with *perfect* and *trustworthy i.i.d.* samples; and ($iii$) big learning enables (implicit) communications among tasks, which is expected to transfer advantages that benefit each other.

## F  EXPERIMENTAL SETTINGS

We employ the same model architectures in the previous Section C for the experiments on the MNIST and CelebA datasets, with the detailed hyperparameters summarized in Table 3. Despite the relatively small models used, we find that big learning is capable of delivering potentially all joint/conditional/marginal data capabilities simultaneously. We adopt the AdamW optimizer (Loshchilov & Hutter, 2017) with $\beta = (0.1, 0.999)$ and constant learning rates for both the generator and the discriminator. Code will be released upon publication.

Table 3: Hyperparameters used in the experiments.

| Dataset | MNIST | CelebA |
|---|---|---|
| Image size | 64 | 120 |
| Patch size | 8 | 10 |
| G-Encoder depth | 6 | 6 |
| G-Encoder #heads | 8 | 8 |
| G-Encoder dim | 256 | 256 |
| G-Decoder depth | 6 | 6 |
| G-Decoder #heads | 8 | 8 |
| G-Decoder dim | 512 | 512 |
| D depth | 6 | 6 |
| D #heads | 8 | 8 |
| D dim | 256 | 256 |
| GP (Mescheder et al., 2018) | real | real |
| $\lambda_{\text{GP}}$ | 10 | 10 |
| Learning rate | $10^{-4}$ | $10^{-4}$ |
| Batch size | 256 | 128 |
| Source ratio $\|\mathbb{S}^1\|/\|\mathbb{L}\|$ | Beta(0.5,3) | Beta(0.5,3) |
| Target ratio $\|\mathbb{T}^1\|/\|\mathbb{L}\backslash\mathbb{S}^1\|$ | Beta(3,0.5) | Beta(3,0.5) |
| Communication source ratio $\|\mathbb{S}^2\|/\|\mathbb{S}^1\cup\mathbb{T}^1\|$ | Beta(0.5,3) | Beta(0.5,3) |

Overall, we find it's quite straightforward to implement the MNIST experiments with the standard implementations discussed in Sections B and C, without resorting to any "tricks" like warm-up or gradient clipping. However, on the more complicated CelebA experiments, we find it's necessary to employ some, as detailed below.

- We employ warm-up in the first 10 epochs for both the GAN generator and discriminator; after that, we use the constant learning rate given in Table 3.

- We apply gradient clipping, with the max norm of 5, to both the generator and discriminator optimizers.

- Similar to Lee et al. (2021), we also find it challenging to stabilize GAN training with a ViT-based discriminator. To deal with that, we additionally ($i$) overlap image patches (Lee et al., 2021) with *e.g.,* 2 pixels at the input of the discriminator (different from the non-overlapping image patches used in the vanilla ViT); and ($ii$) use a larger hyperparameter $\epsilon = 10^{-5}$ in the AdamW optimizer.

Other empirical experiences are listed below.

- We empirically find that the last normalization layers of both the GAN generator and discriminator have a significant influence on the learning stability and final performance. Specifically, replacing the last `LayerNorm` of the G-Decoder of the generator with a `LeakyReLU` leads to improved generative performance, whereas replacing the last `LayerNorm` of the discriminator with other normalization/activation layers results in training collapse.

- Employing an additional convolutional head (like a 3-layer CNN) to the output of the generator often leads to improved performance and training stability.

- Instead of only introducing noise embeddings at the first layer of the G-Decoder of the generator, as shown in Fig. 7a, we find it's beneficial to concatenate the same set of noise embeddings layer-wisely into the G-Decoder layers.

## G   EMPIRICAL EVALUATIONS ON THE GLUE BENCHMARK

Concerning the empirical comparisons between existing methods for foundation models and the presented big learning, intuitively, one would consider first using our big learning as the pretraining strategy in place of existing ones, followed by applying the same naïve fine-tuning on downstream tasks, to evaluate the effectiveness of our big learning. Unfortunately, we cannot afford the pretraining cost; for example, to pretrain a XLNet-Large takes about 5.5 days on **512 TPUs** according to Yang et al. (2019). We leave that to the community, as mentioned in the Conclusion.

To demonstrate the advantages of our big learning over existing methods for foundation models, we alternatively consider leveraging it to serve as the less expensive fine-tuning strategy. It's worth highlighting that, from another perspective, such experiments also verify the advantages of the big learning in the fields of supervised learning, when compared to existing supervised learning methods.

Specifically, we design experiments based on the Hugging Face transformers library Wolf et al. (2020), the GLUE benchmark Wang et al. (2018), and the XLNET Yang et al. (2019) that outperforms the BERT on many NLP tasks. We employ the same pretrained `xlnet-base-cased` model and continually train it on the downstream RTE/MRPC/SST-2 classification tasks via ($i$) the naive fine-tuning (*i.e.,* identical to the original XLNET, termed FT) and ($ii$) our big learning (termed big-learn), respectively. In other words, the pretraining phase (*i.e.,* the permutation language modeling Yang et al. (2019), a special case of our big learning) is the same and we compare our big-learn with the naive FT during the finetuning phase.

Because the data of the downstream classification tasks contain both feature $\boldsymbol{x}$ and label $y$, we resort to the big learning settings of Section 3.3 of the main manuscript. Specifically, $\boldsymbol{X} = (\boldsymbol{y}, \boldsymbol{x})$ and the universal foundation model $p_{\boldsymbol{\theta}}(\boldsymbol{X}_{\mathbb{T}'}|\boldsymbol{X}_{\mathbb{S}'})$ has a network architecture similar to the one shown in Fig. 2 of the main manuscript. Note $p_{\boldsymbol{\theta}}(\boldsymbol{X}_{\mathbb{T}'}|\boldsymbol{X}_{\mathbb{S}'})$ consists of the pretrained XLNET backbone and a task-specific head that is attached to the output of the `<CLS>` token; for simplicity, we abuse $\boldsymbol{\theta}$ to represent all the parameters. For a specific $(\mathbb{S}', \mathbb{T}')$ pair, $p_{\boldsymbol{\theta}}(\boldsymbol{X}_{\mathbb{T}'}|\boldsymbol{X}_{\mathbb{S}'})$ recovers $p_{\boldsymbol{\theta}}(y|\boldsymbol{x})$, *i.e.,* a conventional classifier.

With the above notations, we next formalize the objective for both FT and our big-learn.

- **FT.** Often a cross-entropy is employed, which is identical to

$$\mathcal{L}_{\text{FT}}(\boldsymbol{\theta}) = \mathbb{E}_{q_{\text{downstream}}(\boldsymbol{x}, y)}[-\log p_{\boldsymbol{\theta}}(y|\boldsymbol{x})], \tag{17}$$

  where $q_{\text{downstream}}(\boldsymbol{x}, y)$ represents the training data of the downstream classification task.

- **Big-learn.** For direct comparisons, we formalize the big-learn objective as

$$\mathcal{L}_{\text{big-learn}}(\boldsymbol{\theta}) = \mathcal{L}_{\text{FT}}(\boldsymbol{\theta}) + \beta_{\text{BigLearn}}\mathcal{L}(\boldsymbol{\theta}), \tag{18}$$

  where $\beta_{\text{BigLearn}}$ is a hyperparameter and

$$\mathcal{L}(\boldsymbol{\theta}) = \mathbb{E}_{q(\mathbb{S}', \mathbb{T}')}\mathbb{E}_{q_{\text{downstream}}(\boldsymbol{X})}[-\log p_{\boldsymbol{\theta}}(\boldsymbol{X}_{\mathbb{T}'}|\boldsymbol{X}_{\mathbb{S}'})], \tag{19}$$

  with $q(\mathbb{S}', \mathbb{T}')$ denoting the sampling process of $(\mathbb{S}', \mathbb{T}')$. We simply reuse the same sampling process in Table 3.

Note Eq. (19) is equivalent to minimizing $\mathbb{E}_{q(\mathbb{S}', \mathbb{T}')}\text{KL}[q_{\text{downstream}}(\boldsymbol{X}_{\mathbb{T}'}|\boldsymbol{X}_{\mathbb{S}'})||p_{\boldsymbol{\theta}}(\boldsymbol{X}_{\mathbb{T}'}|\boldsymbol{X}_{\mathbb{S}'})]$ by referring to Eq. (1) of the main manuscript.

Table 4: Tested hyperparameters when comparing FT with big-learn on the GLUE benchmark.

| Task\Hyperparameter | Learning Rate | #Epochs | WarmUp Steps | $\beta_{\text{BigLearn}}$ |
|---|---|---|---|---|
| RTE | [2e-5, 4e-5, 6e-5] | [3, 4, 7, 10, 15] | [0, 120] | [0., 0.2, 0.4, 0.6, 0.8] |
| MRPC | [2e-5, 4e-5, 6e-5] | [3, 4, 7, 10, 15] | [0, 120] | [0., 0.2, 0.4, 0.6, 0.8] |
| SST-2 | [2e-5, 4e-5, 6e-5] | [2, 3, 4] | [0, 1200] | [0., 0.2, 0.4] |

We extensively compare FT with big-learn on the downstream RTE/MRPC/SST-2 classification tasks, by evaluating the accuracy and/or F1 score on the Dev set across the combinations of the tested

hyperparameters shown in Table 4. The hyperparameters are chosen following Devlin et al. (2018); Yang et al. (2019).

Table 5: Empirical evaluations showing the superiority of big-learn to FT. The best/median metrics are calculated among the combinations of the tested hyperparameters in Table 4.

| Metric / Task | Best Accuracy / F1 | | Median Accuracy / IQR | |
|---|---|---|---|---|
| | FT | big-learn | FT | big-learn |
| RTE | 71.84 | **75.09** | 66.06/2.34 | **70.75/1.44** |
| MRPC | 88.97/92.09 | **90.20/93.03** | 87.00/2.45 | **87.74/1.10** |
| SST-2 | 94.15 | **95.18** | 93.75/0.45 | **94.66/0.28** |

The best/median metrics are summarized in Table 5 and Fig. 9 shows the corresponding boxplots; it's clear that our big-learn consistently outperforms FT. Accordingly, our big learning can serve as a superior fine-tuning strategy. It's worth highlighting we did not carefully tune our big-learn; therefore, it's likely that its performance could be further improved by *e.g.,* tuning the sampling process $q(\mathbb{S}', \mathbb{T}')$.

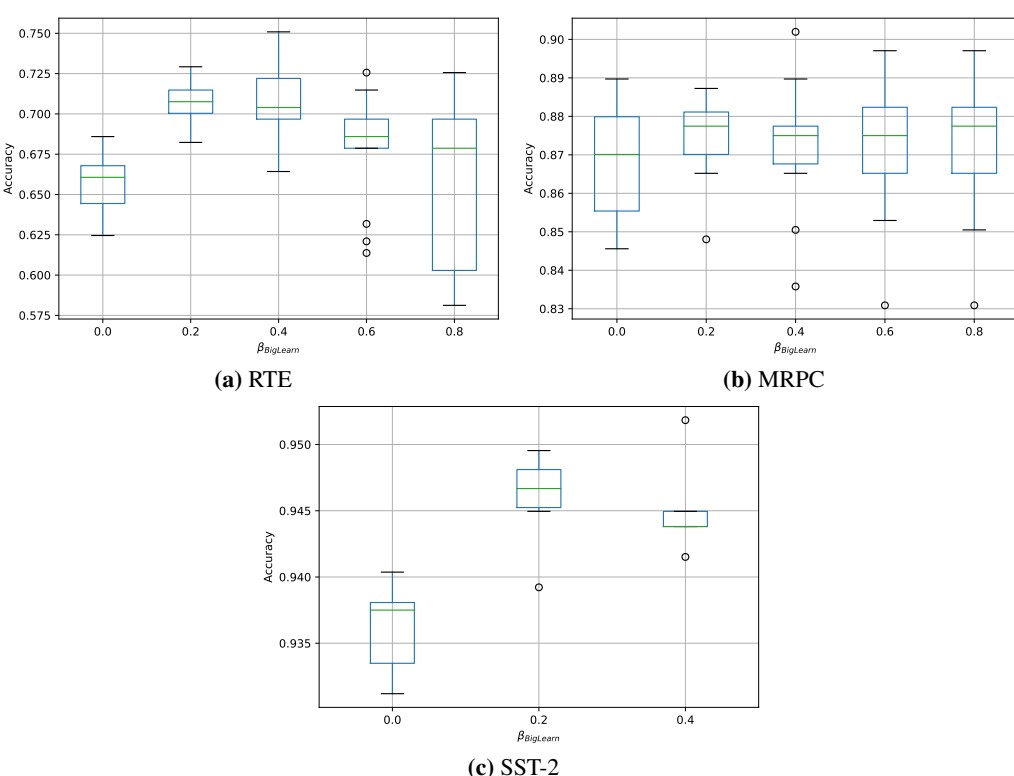

(a) RTE

(b) MRPC

(c) SST-2

Figure 9: Boxplots of the Dev-set accuracies from FT and our big-learn. Note big-learn with $\beta_{\text{BigLearn}} = 0$ is identical to FT (see Eq. (18)). It's clear that big-learn consistently outperforms FT on all three tasks.

We'd like to emphasize that our big learning can reduce the pretrain-finetuning gap because

- it can act as the pretraining and finetuning objectives, simultaneously;
- one can even rely on our big learning to completely merge the pretraining and finetuning phases, leading to a zero gap.

Motivated by the performance boost from the BERT to the XLNET and our discussions "on the generalization of model parameters and latent features" of Section 3.2 of the main manuscript, we posit that our big learning can serve as better pretraining and finetuning strategies than existing

methods, leading to a universal machine learning paradigm. We leave the corresponding verification as future research.

# H  ADDITIONAL EXPERIMENTAL RESULTS

More experimental results, complementing the limited demonstrations of the main manuscript, are given below. Please refer to the captions for details.

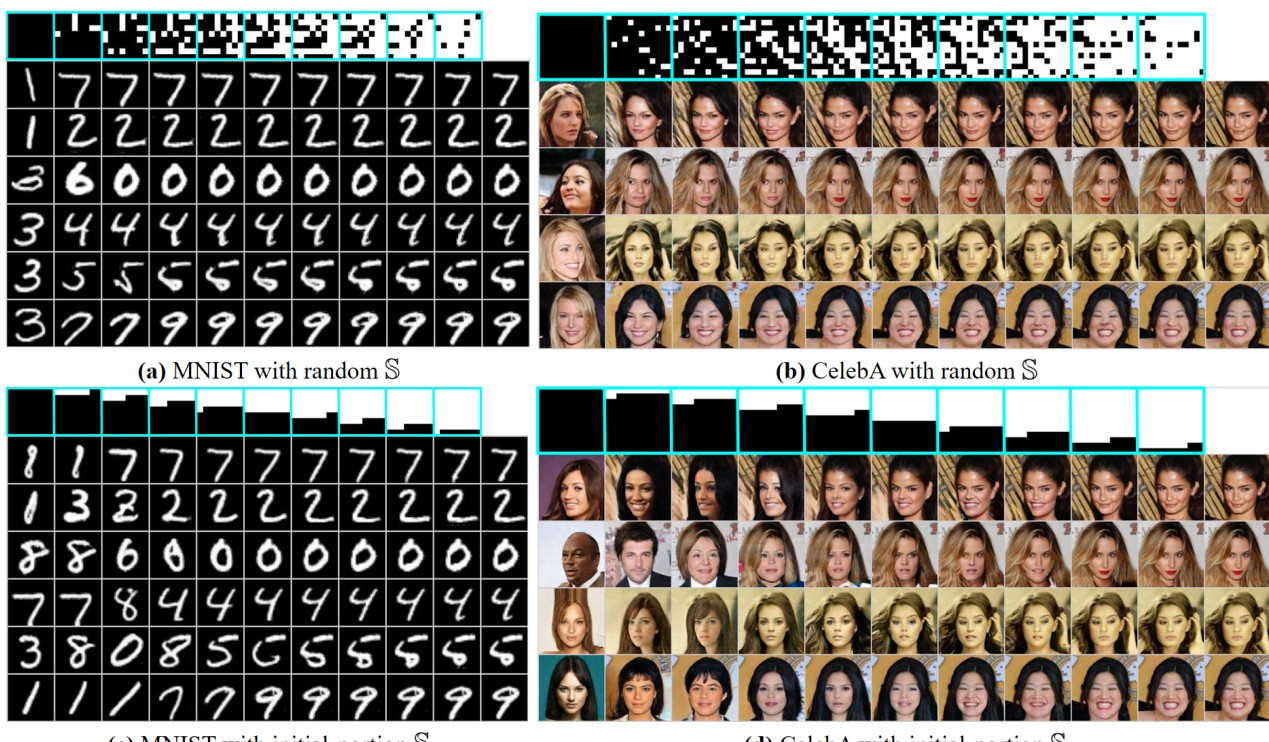

**(a)** MNIST with random $\mathbb{S}$

**(b)** CelebA with random $\mathbb{S}$

**(c)** MNIST with initial-portion $\mathbb{S}$

**(d)** CelebA with initial-portion $\mathbb{S}$

Figure 10: Demonstrating the generation/completion capabilities of big learning when gradually increasing the ratio of $\mathbb{S}$ from 0 (joint generation) to 0.9, from left to right. Shown in the light-blue boxes of the first row are the masks of $x_{\mathbb{S}}$ applied in each column; white/black indicates $\mathbb{S}/\mathbb{T}$. The right-most column shows ground-truth $x$ shared in each row. Note each row also employs the same noise. It's clear that the generations become increasingly similar/dissimilar to the ground-truth $x$ as the ratio of $\mathbb{S}$ increases/decreases, as expected. See the category, style, and thickness of the MNIST generations as the ratio of $\mathbb{S}$ decreases, as well as the identity, expression, hairstyle, and gender of the CelebA generations. Big learning produces realistic and diverse generations/completions in all situations.

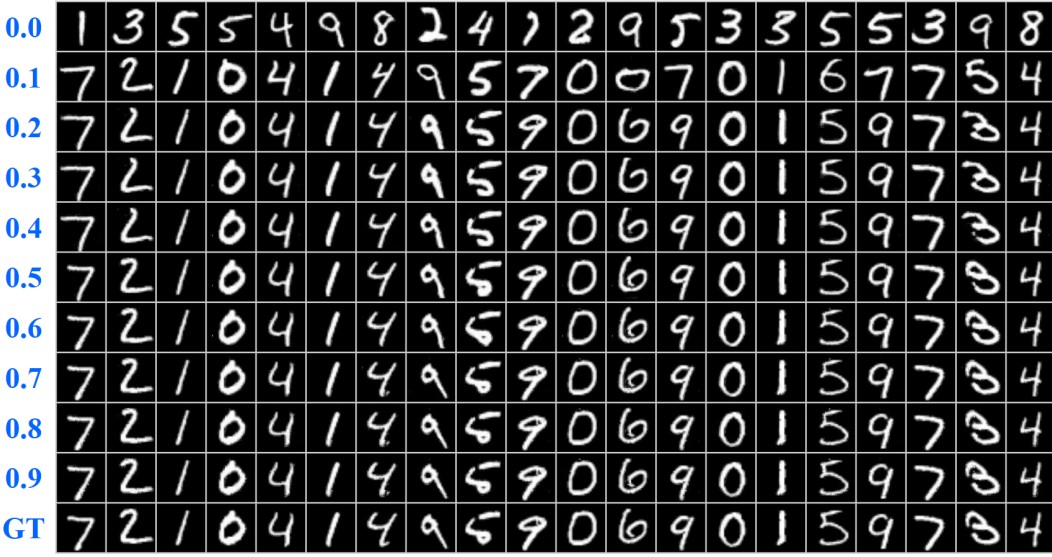

Figure 11: More MNIST generations/completions from big learning when gradually increasing the ratio of $\mathbb{S}$ from 0.0 to 0.9.

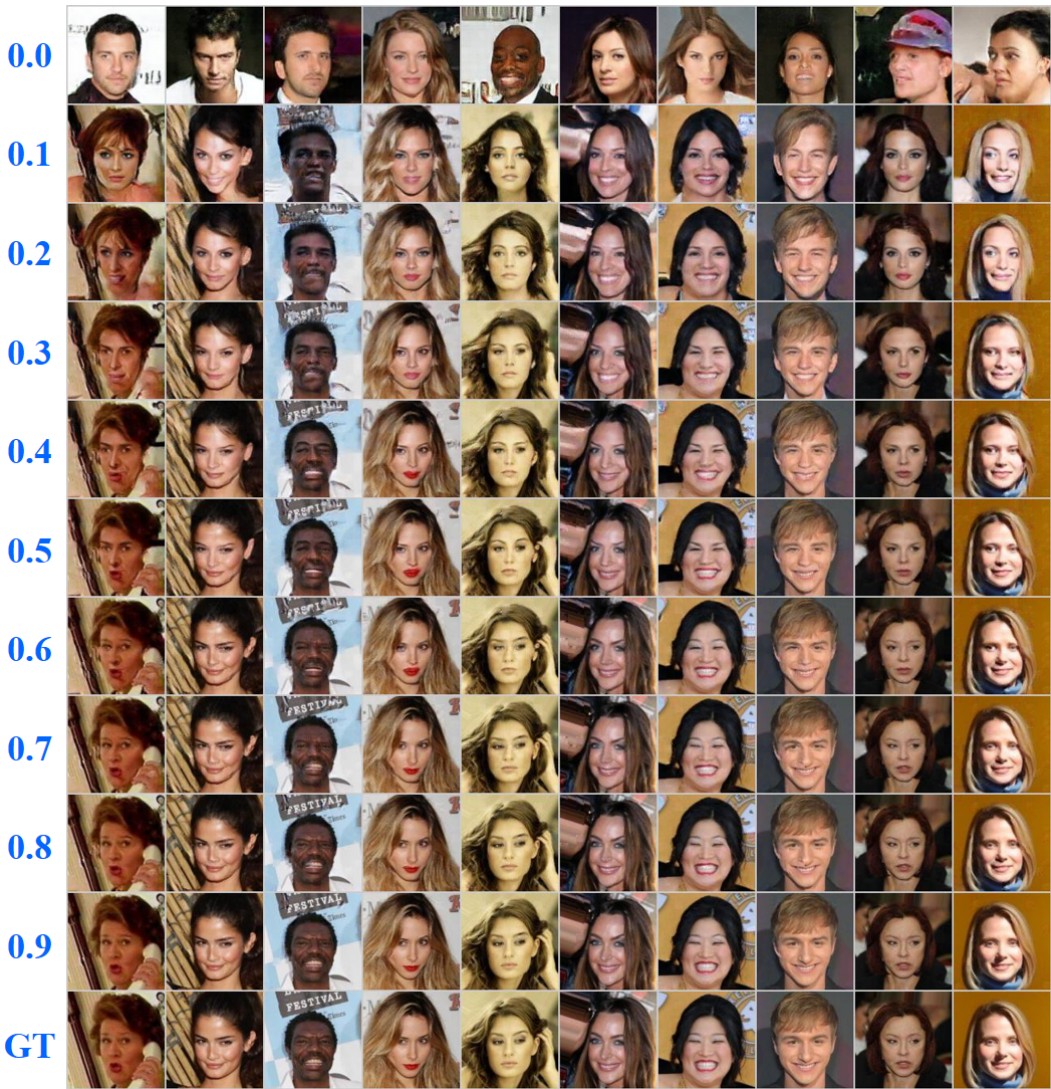

Figure 12: More CelebA generations/completions from big learning when gradually increasing the ratio of $\mathbb{S}$ from 0.0 to 0.9.

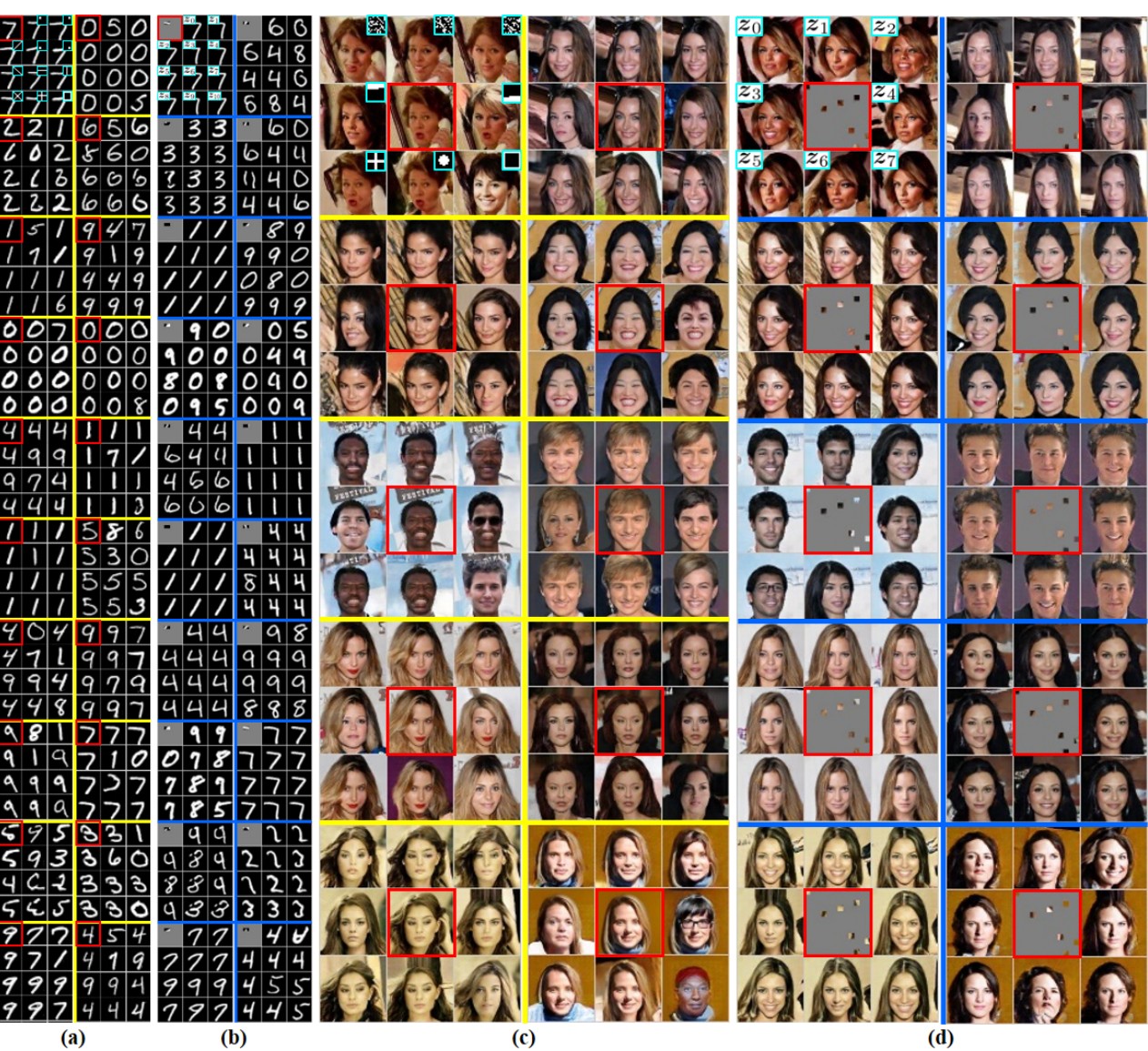

Figure 13: The diverse generations/completions of big learning with (a)(c) various $\mathbb{S}$ settings and (b)(d) different noises. Shown in red boxes are either the ground-truth images $\boldsymbol{x}$ or the source $\boldsymbol{x}_{\mathbb{S}}$. Big learning delivers diverse realistic generations *w.r.t.* different $\mathbb{S}$/noise settings.

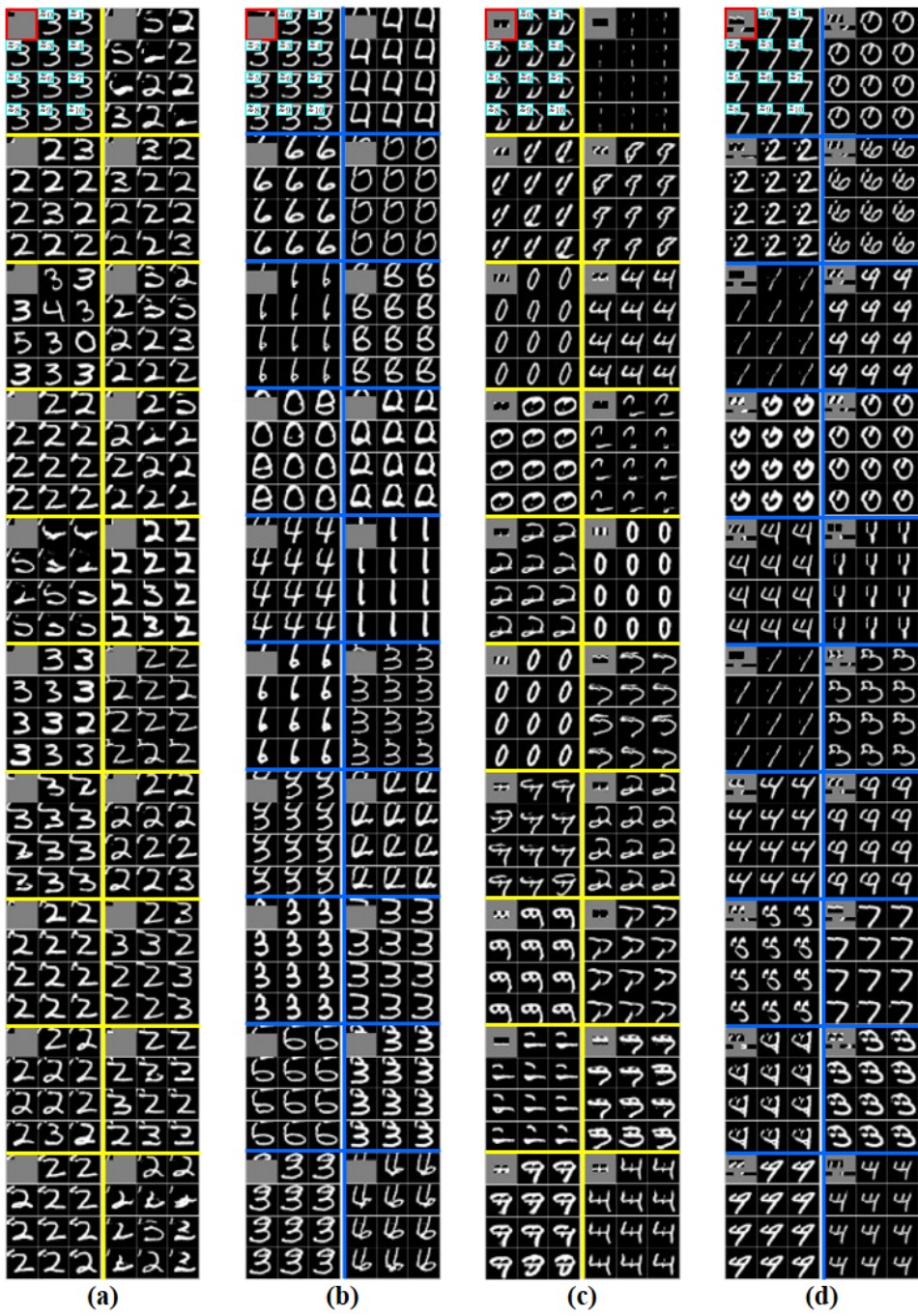

Figure 14: The strong generalization capability of big learning *w.r.t.* anomalous testing cases out of the training domain. Big learning generalizes well on $x_{\mathbb{S}}$s that are constructed with (a) random center patches replaced in the upper-left corner, (b) random center patches replaced in the upper part, (c) random center patches duplicated and replaced in the center, and (d) random patches and more complicated manipulations (including duplication, relocation, and mix-up).

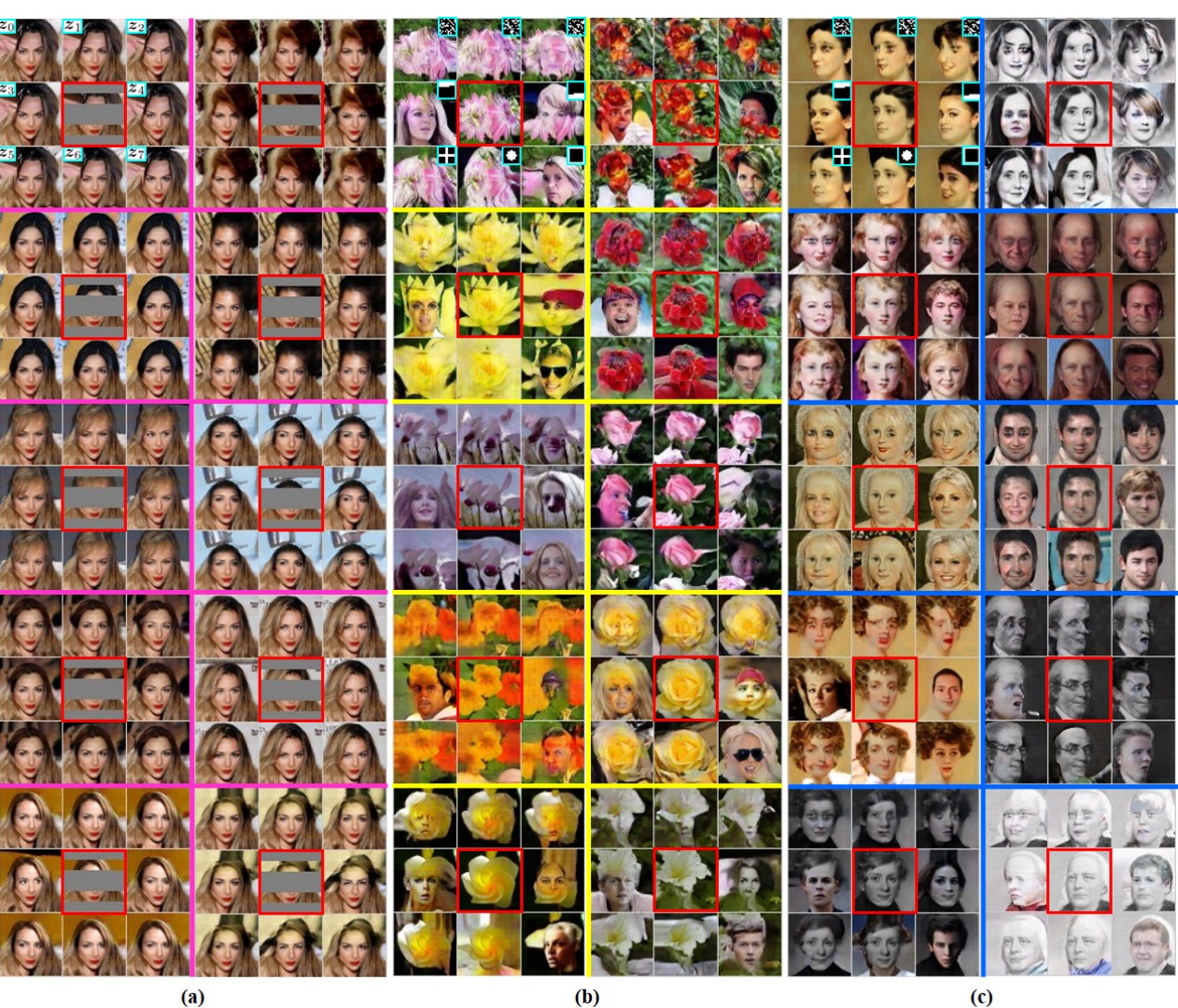

(a)                                        (b)                                        (c)

Figure 15: The strong generalization capability of big learning *w.r.t.* anomalous/unseen testing cases out of the training domain, on (a) CelebA, (b) Flowers, and (c) MetFaces. Big learning generalizes well on $x_{\mathbb{S}}$ constructed by (a) mixing-up patches from different CelebA images, (b) sampling out-of-domain image patches from the Flowers dataset, and (c) sampling out-of-domain image patches from the MetFaces dataset.

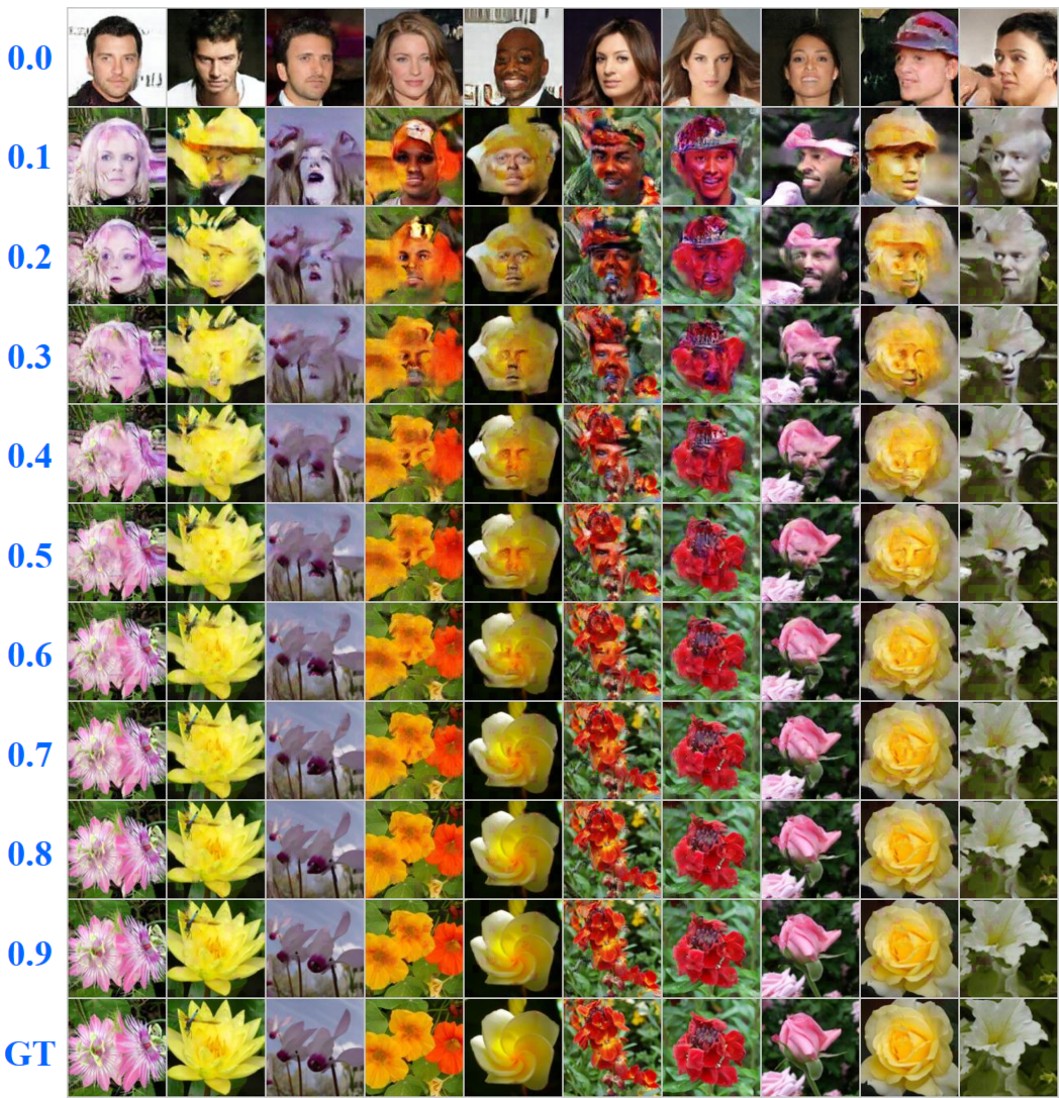

Figure 16: Out-of-domain generations/completions from big learning on the Flowers, when gradually increasing the ratio of $\mathbb{S}$ from 0.0 to 0.9. The tested model is big-learned on the CelebA.

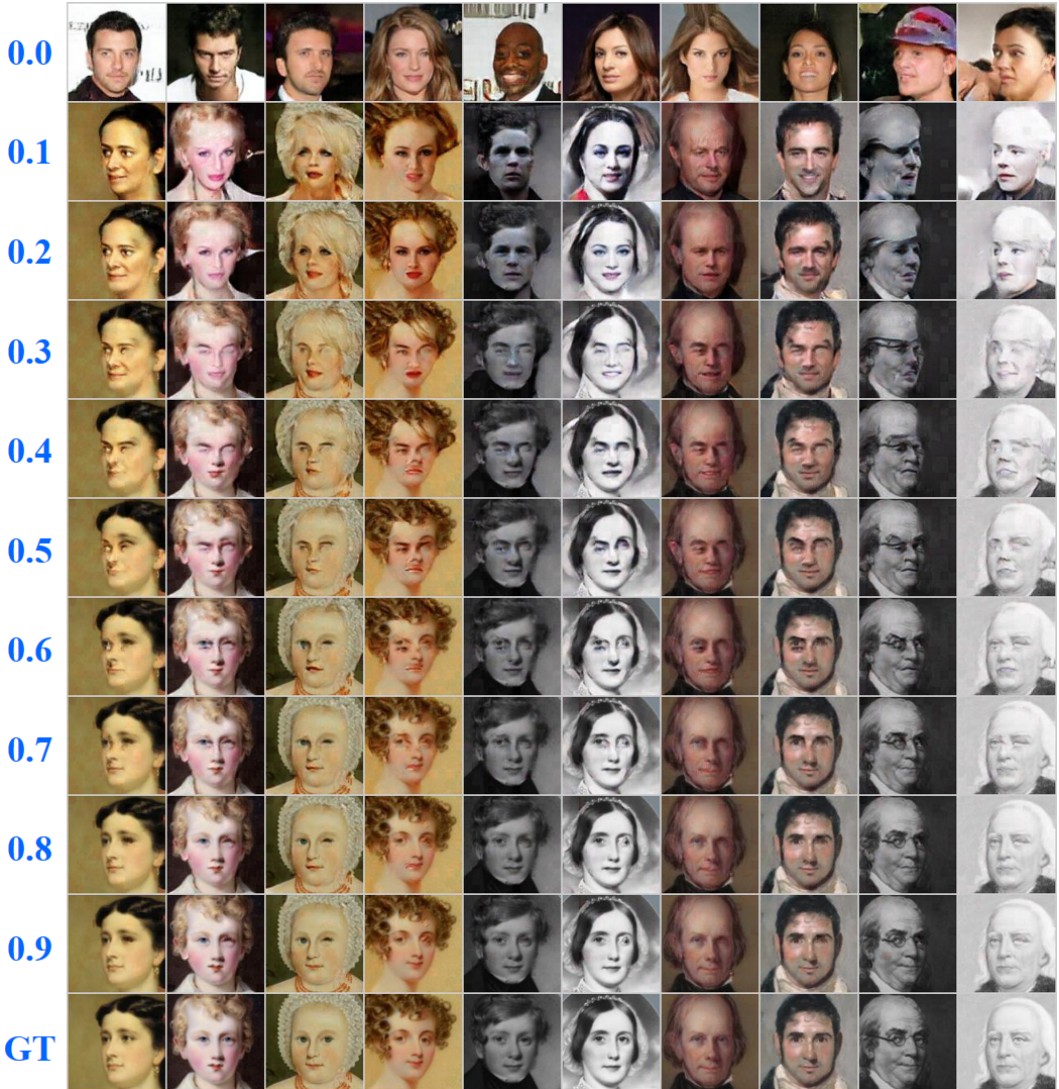

Figure 17: Out-of-domain generations/completions from big learning on the MetFaces, when gradually increasing the ratio of $\mathbb{S}$ from 0.0 to 0.9. The tested model is big-learned on the CelebA.

