# OpenReview forum: "Big Learning: A Universal Machine Learning Paradigm?"
_ICLR.cc/2023/Conference — Submitted to ICLR 2023_

### Official Review · Reviewer_hoLo · 2022-10-20

**Confidence:** 2
**Correctness:** 2
**Technical Novelty And Significance:** 3
**Empirical Novelty And Significance:** 2
**Recommendation:** 5

**Clarity, Quality, Novelty And Reproducibility:**

The paper presents a novel method as it aims to improve foundation models such as BERT by providing a theoretical framework that explains the success of the foundation models. The authors do not share the code and data used in the experiments. As a result, this may be difficult to reproduce.

**Strength And Weaknesses:**

Strength: The mathematical descriptions are well defined. Also, the proposed Big learning method has an advantage of handling incomplete data and missing values.

Weakness : The abstract is poorly written, ambiguous,  and does not describe the problem. For example, the statement “ Recent breakthroughs based on big/foundation models reveal a vague avenue for AI, that is, big data, big/foundation models, big learning, · · · .”. Exactly what avenue is being explored here? Also, decide on a single term to describe the problem and not include multiple terms such as “big data, big/foundation models, big learning”.  This leaves a lot of room for confusion and misinterpretation.  This happens throughout the rest of the paper as well.

**Summary Of The Paper:**

This work provides a new method named “Big learning” that provides a theoretical platform for analysing, justifying, and improving foundation models such as BERT.

**Summary Of The Review:**

This paper presents a new method named Big Learning that attempts to improve upon popular NLP models such as BERT by exploiting the information inherent in large scale data through modelling marginal data distributions with a single universal model.

Overall, although the proposed method is novel,  the paper is poorly written ambiguous as the problem is not simply and clearly defined. Several different words are used to describe the same thing which might lead to confusion and misinterpretation by the reader. Although the mathematical expressions are clearly defined, this gets lost in the unclear text.

---

> ### Author Response · Authors · 2022-11-18
> **Responses to REVIEWER HOLO**
>
> `Q1: “Exactly what avenue is being explored here? …This happens throughout the rest of the paper as well.”`
>
> A1: It’s impossible to clearly explain everything in the Abstract; in the Introduction (specifically the first paragraph
> of Page 2), we explain the vague AI avenue in detail, i.e., “…one leverages \emph{big data} to comprehensively represent
> the underlying data distribution, develops \emph{big/foundation models} to serve as a big information ``container,''
> relies on \emph{big learning} to comprehensively and exhaustively convey data information into that container, and so
> on.”
>
> Perhaps the following rephrasing will help, i.e., “… a vague avenue for AI, whose roadmap covers big data,
> big/foundation models, big learning, …”
>
> The terms `big data` and `big/foundation models` are well-known to the community, so there should not be confusion and
> misinterpretation, right? As for the newly introduced big learning, the whole paper is for its explanation. It’s reasonable that a newly
> introduced method is not crystal clear in the Abstract of the paper, right?
>
> Please also constructively suggest where is the “confusion and misinterpretation?” in the rest of our paper. Thanks.
>
> `Q2: “The authors do not share the code and data used in the experiments. As a result, this may be difficult to
> reproduce.”`
>
> A2: We followed the ICLR submission protocols. Please see our responses “On the code,” run our code, and revise your
> comments correspondingly. Thanks.
>
> `Q3: “… the problem is not simply and clearly defined. … Several different words are used to describe the same thing …
> unclear text”`
>
> A3: We are quite confused. In your comments on the advantages of our paper, you mentioned “the mathematical descriptions are
> well defined,” but you also contradictorily commented, “the problem is not simply and clearly defined.” Please elaborate
> on your concerns, which we very much want to address. Looking forward to your reply.
>
> Please constructively give some examples of “Several different words are used to describe the same thing” and “the
> unclear text.” We’d like to address any misunderstanding. Thank you.

---

### Official Review · Reviewer_qcrZ · 2022-10-24

**Confidence:** 4
**Correctness:** 2
**Technical Novelty And Significance:** 1
**Empirical Novelty And Significance:** 1
**Recommendation:** 1

**Clarity, Quality, Novelty And Reproducibility:**

Clarity
The paper is quite readable, but the arguments presented are not clear. Many points are made without sufficient explanation or proof. Many comparative sentences just say it is “more” or “better”. Better than what? Concretely, theoretically, empirically, how is it better and what is it better than? There is a lot of language but not sufficient concrete evidence to support the claims made.

Quality
The paper has typos, and because of the poor evidence for the claims fails to meet a high quality standard.

Novelty
The paper claims extreme novelty, but does not justify it. The current presentation does not seem novel.

Reproducibility
I have not gone through the appendices in full detail, but experimental details are provided along with more discussion of some points made in the main paper.
No code is provided to replicate the results presented in the experiments section.

**Strength And Weaknesses:**

Strengths
1) The paper approaches an interesting idea. Understanding learning problems as specific instantiations of modeling the complete joint distribution over data is a compelling goal, and the basic formulation of learning all P(S|T) for all subsets S,T makes intuitive sense.

Weaknesses
1) The paper ultimately fails to prove the value of its proposition. The proposed framework is never explicitly defined, and seems to have a varying definition to fit the argument being made at any point in the text. It appears to encompass "all possible probability distrubtion modelling", but its not clear how this generalization is valuable.

a) 3.2 Discussions has a lot of strong claims that are not well substantiated.
how exactly can you share one model? this needs to be explicit: how does p_\theta(x_T|x_S) learned for all S,T yield all joint/conditional/marginals that are close to q(x_T|x_S) for a specific S,T? The empirical argument is fine but does not justify this new “big learning” paradigm unless it provides some new theoretical advantage or insight.

b) Why or how does the framework generalize past the transformer/ViT models? Because they are “data/information hungry” is odd, it's not clear what the authors mean by this. Not enough evidence is provided that “big learning” generalizes past the transformer/ViT/BERT models presented.

c) “model capacity likely not an issue” the discussion here seems unnecessary if the assumption is that the model is "big enough".

d) How is this different from simple masking or conditional generation? what new insight should a reader be getting?

e) weighting massive data tasks:
the optimum is the same regardless of different weighting strategies? how? The problem is still highly nonconvex, how is it guaranteed that the optimum will be the same if the training procedure is different via different sampling?

f) Eq.6 looks like it is exactly making some assumption about the factorization of the distribution. How does this fit into the model of learning all joint/conditional/marginals? If I wanted to sample from p_\theta(X_T’|y_S’), how would I do that?


g) How is this different from masking/conditional generation that already exists with different mask levels? All experiments seem to be showing various masking results with no comparison to other frameworks, and do not seem to provide much new observations that are only possible because of “big learning”.


Notes/Other

1) The structure of the arguments are difficult to follow. It's hard to see what is a technical result and what is just discussion. It would benefit significantly from definition/theorem/proposition blocks.

2) Figures 1 and 2 do not add much over the text, and table 1 is very vague and not well substantiated.

**Summary Of The Paper:**

The paper proposed "big learning", a framework for understanding and modeling complex probability distributions. Big learning is defined to be a modeling of all possible factorizations of a given probability distribution, with all conditionals, marginals, and joints fully learned. The paper shows that existing modeling frameworks fit under the umbrella of big learning as special cases. Experiments with masking show compelling conditional generation results.

**Summary Of The Review:**

The paper presents a new framework, but fails to demonstrate its value, both theoretically and experimentally. Many arguments are described with text, but ultimately are not proven or substantiated.

---

> ### Author Response · Authors · 2022-11-18
> **Responses to REVIEWER QCRZ (1/2)**
>
> `Q1: “…The proposed framework is never explicitly defined, … but its not clear how this generalization is valuable.”`
>
> A1: The proposed framework was explicitly defined in the paragraph with Eq. (3) on Page 4, where the random variable $x$
> may represent a continuous image or a discrete text.
>
> Our big learning contains many foundation models (across many
> domains like CV/NLP) as special cases and unifies them into a principled intuitively-simple framework in
> Eq. (3). That generalization of big learning and its intuitive simplicity are the main advantages (not disadvantages) of
> the presented big learning. Please refer to our responses “On understanding our big learning” and “On comparisons with
> existing methods.”
>
> Our big learning also has the outstanding advantage of flexible implementations, where one can flexibly specify maximum
> likelihood learning, adversarial learning, or even diffusion models to implement Eq. (3); see our discussion on “Model
> architecture and training objective” following Eq. (3). It’s therefore impossible and not reasonable to formalize it
> into one specific loss function. That implementation flexibility is of great value concerning versatile practical
> applications.
>
> We further contribute by proposing a novel adversarial implementation example in Eqs (4) and (5).
> Please refer to our responses “On comparisons with existing methods” and “On the (downstream) use/value of big learning
> and big-learned capabilities” for the answer to your question “how this generalization is valuable.”
>
> `Q2: “… how exactly can you share one model? … how does p_\theta(x_T|x_S) … close to q(x_T|x_S) for a specific S,T? …”`
>
> A2: Please see Eq. (3) and Fig. 1(a), where all $p_{\theta}(\cdot)$s are modeled with a universal Transformer/ViT, with the
> same set of parameters $\theta$; for a specific S,T pair, we follow Fig.1(a) to leverage that universal Transformer/ViT to
> model the generative process of $p_{\theta}(x_T|x_S)$, where $x_S$ is the input and $x_T$ is the output (see Eqs. (4) and (5)
> and Appendix C for details).
> As Eq. (3) suggests, for a specific S,T pair/task, big learning trains $p_{\theta}(x_T|x_S)$ to approximate the corresponding $q(x_T|x_S)$.
> Besides, ideally, Eq. (3) holds true for all S,T pairs, as discussed in the first discussions of Section 3.2.
> Accordingly, the big-learned universal Transformer/ViT yields all joint/conditional/marginal data capabilities after training.
>
> For $x$ denoting a sequence of discrete tokens, please refer to the third paragraph of Page 5. In this case, one can use
> a universal Transformer/ViT to directly model all $p_{\theta}(\cdot)$s, with the same main idea of Eq. (3) and Fig. 1(a).
>
> `Q3: “… unless it provides some new theoretical advantage or insight …” “How is this different from masking/conditional
> generation…new observations that are only possible because of “big learning” `
>
> A3: We have made non-trivial contributions both theoretically and empirically. Please see our responses “On comparisons
> with existing methods” and our answers to the Q2 of Reviewer ZM7W. Should you have any newly raised questions, please do
> share them with us. Thanks.
> Please also refer to the following QA for additional information.
>
> `Q4: “Why or how does the framework generalize past the transformer/ViT models? … How is this different from simple
> masking or conditional generation? what new insight should a reader be getting?”`
>
> A4: Our contributions are associated with the learning of foundation models; it’s orthogonal to the architecture design
> of transformer/ViT models. Please see our responses “On comparisons with existing methods,” where it’s crystal clear
> that our big learning generalizes existing heuristic learning methods (like the mask-and-predict) for foundation
> models (like BERT) with a lot of advantages/improvement/new-insights.
>
> When comparing our big learning to the mask-and-predict, we’d like to highlight four example
> advantages/insights:
> - big learning can naturally handle complete/incomplete data;
> - big learning exhaustively models the correlations among the predicted $x_T$, which are often ignored by the
> mask-and-predict;
> - our big learning reveals what the heuristic mask-and-predict learning is actually doing, i.e., simultaneously
> modeling many conditional data distributions; and
> - we propose the adversarial implementation in Eqs. (4) and (5), which are clearly novel and beyond the scope
> of existing mask-and-predict learning.
>
> `Q5: “… data/information hungry” is odd, it's not clear what the authors mean by this.”`
>
> A5: As clearly stated in the last line of Page 2, Transformers/ViTs are well-known to be over-parameterized with a lot of
> redundancy within their parameters. That means there are plenty of “rooms” for data information within
> Transformers/ViTs. Accordingly, Transformers/ViTs are often considered to be “data/information hungry”; e.g., [1,2] use
> the same terminology.

---

> ### Author Response · Authors · 2022-11-18
> **Responses to REVIEWER QCRZ (2/2)**
>
> `Q6: “weighting massive data tasks: the optimum is the same…how? …”`
>
> A6: The (global) optimum is determined by $q(x)$, not the modeling of $p_{\theta}(x)$. Note we assume sufficiently large
> model capacity of $p_{\theta}(x)$.
>
> Please refer to our discussion on “Can we share one universal foundation model …”, specifically, “all
> conditional/marginal data distributions $q(x_{T}|x_{S})$ can be derived from the joint one $q(x)$, meaning
> that their perfect modelings should share the same set of parameters” $\theta^*$. Accordingly, $\theta^*$ is the (
> global) optimum, regardless of different weighting strategies for S,T.
>
> Different weighting strategies may lead to different (local) optima. However, that global optimum, defined by $q(x)$, is
> always one of them and stays the same, regardless of different weighting strategies for S,T.
>
> `Q7: “Eq.6 making some assumption about the factorization of the distribution. …learning all
> joint/conditional/marginals?... sample from p_\theta(X_T’|y_S’), how would I do that?”`
>
> A7: Our goal here is to highlight the advantage of recursive usage of the universal model that naturally comes with our big
> learning. Our big learning makes no assumption on the factorizations of data distribution in general; we will emphasize
> that in the revised manuscript to remove the rare ambiguity. Note, considering practical applications, one can of course
> use a special case of our big learning, e.g., to simultaneously model many (not all) joint/conditional/marginals.
>
> To sample from $p_{\theta}(X_{T’}|y_{S’})$, simply add the corresponding task to the big learning. Without loss of generality, let’s
> assume $S’=\emptyset$, and $X_{T’}={x,y}$ for simplicity, where $x$ is continuous and $y$ is discrete following the settings of
> Eq (6). Then, the LHS of Eq. (6) can be rewritten as $p_{\theta}(x,y)$, i.e., a joint distribution of continuous $x$ and
> discrete $y$. Eq. (6) shows one factorization of $p_{\theta}(x,y)= p_{\theta}(y|x) p_{\theta}(x)$, where the gradient wrt $\theta$
> may be readily calculated, thanks to the continuity of $x$. Should sampling from $p_{\theta}(x|y)$ be of interest, one can of
> course form the other factorization, $p_{\theta}(x,y)= p_{\theta}(x|y) p_{\theta}(y)$, into another learning task of our big
> learning, despite one has to deal with the challenge of backpropagation through the discrete $y$. Potential solutions
> include
> - to resort to the Gumbel-SoftMax technique, and
> - to separately and approximately train $p_{\theta}(x|y)$ and $p_{\theta}(y)$ without considering the backpropagation through discrete $y$.
>
> The above discussions will be added to the Appendix, for the sake of the clarity of the main manuscript.
>
> `Q8: “… Many points are made without sufficient explanation or proof. Many comparative sentences just say it is “more” or
> “better”. Better than what? … how is it better and what is it better than …”`
>
> A8: Would you please constructively give some examples? Thanks. We revised the original submission carefully; there should
> not be confusions based on the context.
>
> `Q9: “Quality … Novelty…. Reproducibility …No code is provided…”`
>
> A9: Please refer to our above responses for your concerns on Quality and Novelty.
> We followed the ICLR code submission protocols. Please see our responses “On the code,” run our code, and revise your
> comments correspondingly.
>
> Please share with us your newly-raised concerns, which we very much want to address. Thanks.
>
> [1] Hassani, A., Walton, S., Shah, N., Abuduweili, A., Li, J., & Shi, H. (2021). Escaping the big data paradigm with
> compact transformers. arXiv preprint arXiv:2104.05704.
>
> [2] Wang, W., Zhang, J., Cao, Y., Shen, Y., & Tao, D. (2022). Towards data-efficient detection transformers. arXiv
> preprint arXiv:2203.09507, 1(3), 7.

---

### Official Review · Reviewer_ZM7W · 2022-10-31

**Confidence:** 4
**Correctness:** 2
**Technical Novelty And Significance:** 2
**Empirical Novelty And Significance:** 2
**Recommendation:** 3

**Clarity, Quality, Novelty And Reproducibility:**

Clarity: The papers goals could be stated more clearly
Quality: The paper does not benchmark any results or provide clear downstream applications
Novelty: The experiments in the paper employ existing methods with some modifications

**Strength And Weaknesses:**

Strengths:
1. The paper reviews various training objectives in literature and shows how they may be restated under their ‘big learning paradigm’.
2.  They conduct experiments on image completion that produce good images completions (though they do not provide any metrics to compare with existing results).

Weakness:
1. The main goal of the paper is unclear. Are the authors proposing a new theoretical paradigm to analyze foundation models? If yes, there are no stated proposals for how their phrasing gives us more insight into existing methods than the existing phrasing. Are they proposing a new experimental paradigm? Is yes, the authors need to clearly state what the downstream goal and use of their experimental paradigm is.
2. Unsubstantiated claims:
- The authors claim that their big learning setup gives you more downstream applications than joint modeling (table 1), but they do not state any particular way to test this claim. For example, in their conditional joint modeling experiments, they show that you can generate images from many different ratios of missing pixels, but they don’t show what the downstream use of this capability is? For instance, they don’t compare to joint modeling methods to show that this method learns better representations for classification.
- “potentially delivers all joint/conditional/marginal data capabilities after training” it is not at all obvious that a model trained on many diverse datasets will be able to model all distributions and this has not been shown via experiments in their paper.
- The paper makes some vague claims like “big learning behavior closely resembles the fundamental unconscious mind and the vision system of human brains”.
- They claim that “big learning is what foundation models are implicitly doing” but again they don’t lay out any clear testable hypothesis for this claim. Which behavior exactly are they referring to, and why is big learning any more explanatory than the loss function of existing foundation models like autoregressive modeling.


Minor comments.
1. Table 1 is unclear. For eg: what do authors  ean that ‘capabilities after training’ are ‘joint’? What is the basis to claim that downstream applications are limited for joint modeling vs. abundant for unsupervised big learning?
2. The paper does not provide any metrics for measuring the quality of results in their paper.


**Summary Of The Paper:**

The paper attempts to unify many different machine learning models by re-stating them as ‘big learning’. The authors claim that this paradigm is better than existing methods since it can handle incomplete data. They conduct experiments on conditional image generation.



**Summary Of The Review:**

The contribution of this paper is unclear - while they attempt to unify existing methods under their big learning paradigm, they do not demonstrate concrete theoretical or practical benefits of doing so. The paper also contains many unsubstantiated claims.

---

> ### Author Response · Authors · 2022-11-18
> **Responses to REVIEWER ZM7W (1/2)**
>
> `Q1: “… produce good images completions (though they do not provide any metrics to compare with existing results).”`
>
> A1: Please refer to our responses “On comparisons with existing methods,” where we quantitively prove that our big
> learning can serve as a superior fine-tuning strategy. Should the reviewer be concerned about the image completion quality
> when compared with existing methods, we emphasize that there are NO such existing methods that are capable of
> simultaneously modeling many/all joint/conditional/marginal data distributions, to our knowledge. Please name one so
> that we can make comparisons. Thanks.
>
> `Q2: “The main goal of the paper is unclear. Are the authors proposing … and use of their experimental paradigm is.”`
>
> A2: Please first refer to our responses “On understanding our big learning.” As the title states, our main goal is to
> introduce the big learning that potentially serves as a universal machine learning paradigm. We make nontrivial
> contributions by revealing existing heuristic learning methods for foundation models (like the mask-and-predict) can be
> unified by our principled intuitively-simple big learning framework; we also contribute by revealing many of its good
> advantages, e.g., dealing with incomplete data, yielding arbitrary data completion, etc. That revealed unification and
> advantages of our big learning are clearly `new “insights.”` Please refer to our responses “On comparisons with existing methods”
> and “On the (downstream) use/value of big learning and big-learned capabilities” for “the downstream goal and use” of
> our big learning.
>
> `Q3: “Unsubstantiated claims: … more downstream applications than joint modeling … what the downstream use of this
> capability is? … joint modeling methods to show that this method learns better representations for classification.”`
>
> A3: The reviewer misunderstands the definition of “joint modeling,” which is formally defined in the first paragraph of
> Section 3.1. By “joint modeling” we mean constructing a joint model $p_{\theta}(x)$ to resemble the joint data
> distribution $q(x)$. Note joint modeling methods (like GANs and GPTs) are rarely used for pretraining representations
> for downstream classification tasks. With that clarified, it’s clear that our big learning (with many methods for
> foundation models as special cases) gives more downstream applications than joint modeling; this is also expected to
> address your minor comments in Table 1 of our paper.
>
> Please see our responses “On the (downstream) use/value of big learning and big-learned capabilities” for the “use of
> this (joint/conditional/marginal data) capability.”
>
> `Q4: “… it is not at all obvious that a model trained on many diverse datasets will be able to model all distributions
> and this has not been shown via experiments …”`
>
> A4: We boldly guess that by modeling “all distributions on many diverse datasets” you imply modeling the digital world with
> many modalities. That is indeed the ultimate use of (the future versions of) our big learning. However, we cannot
> empirically prove the feasibility by ourselves, but we have faith in our community.
> To alleviate your concerns, we highlight the following three facts.
> - As special cases of our big learning, many successful foundation models from diverse research fields have extensively
> verified the feasibility of our big learning from many perspectives.
> - We have empirically verified with the MNIST/CelebA datasets the feasibility of leveraging `one model` to simultaneously model all joint/conditional/marginal PDFs of `one underlying complete data distribution` q(x).
> - For simplified settings with two diverse datasets of texts and images, the existing special case of our big
> learning, i.e., the Dall-E, has partially demonstrated the corresponding feasibility of generating images conditioned
> on texts; we are working on the experiments and will provide the proofs if time permits.
>
> Please see the first discussion of Section 3.2. Based on that discussion, the feasibility of modeling joint/conditional/marginal PDFs with one model is directly determined by the existence of the underlying joint/complete distribution $q(x)$ and a sufficiently large modeling capacity of $p_{\theta}(x)$. Note a distribution can have many modes. Ideally, one can of course significantly expand the scope of $q(x)$ and use it to represent the digital world.

---

> ### Author Response · Authors · 2022-11-18
> **Responses to REVIEWER ZM7W (2/2)**
>
> `Q5: “… vague claims like “big learning behavior closely resembles the fundamental unconscious mind and the vision system
> of human brains” `
>
> A5: Please refer to the cited papers following that claim, where it’s shown that the unconscious mind and the vision system
> of human brains are excellent at comprehensive information exploitation in a multitasking manner. That multitasking
> information exploitation is quite similar to the exhaustive joint/conditional/marginal modeling of our big learning.
> Therefore, we made the corresponding claim to draw connections to the cited papers. Please constructively point out
> where and how the claim is improper so that we can modify it correspondingly. Thanks.
>
> `Q6: “… big learning is what foundation models are implicitly doing … clear testable hypothesis for this claim.”`
>
> A6: We revealed in the paper that many existing heuristic learning methods for foundation models are special cases of our
> intuitively simple big learning. Accordingly, we stated, “common (not all; see Appendix D) foundation models are
> implicitly doing big learning.” We have provided the detailed proofs in “Model architecture and training objective” of
> Section 3.1, “Model architecture and training objective” of Section 3.3, and “BERT pretraining as a special case of big
> learning” of Section 3.3. We have explained in detail how our big learning reduces to existing foundation models, like
> MAE, GPTs, BERT, XLNET, Dall-E.
> Please also refer to our responses “On comparisons with existing methods.”
>
> `Q7: “The experiments in the paper employ existing methods with some modifications.”`
>
> A7: We strongly disagree with your comment.
>
> First of all, to our knowledge, there are no methods that ($i$) unify most existing learning
> methods for foundation models within one framework and ($ii$) simultaneously model many/all joint/conditional/marginal data distributions
> with both complete and incomplete data. Should the reviewer be aware of such an existing method, please constructively
> share its name, as support of this comment.
>
> Secondly, please provide a reference that developed Eqs. (4) and (5), which are our key experimental implementations.
>
> Thirdly, please refer to our responses “On comparisons with existing methods” and challenge our statements therein with proofs.
>
> Please be responsible and review our paper rigorously and objectively. We have spent a lot of effort on it.
>
> `Q8: “The contribution of this paper is unclear … concrete theoretical or practical benefits of doing so …”`
>
> A8: We have listed our main contributions in the Introduction. Please refer to our detailed responses above and share
> with us your newly raised questions if any. Thanks.

---

### Author Response · Authors · 2022-11-18
**We thank all reviewers for their effort in reviewing our paper and in discussing their concerns with us during the rebuttal.**

We thank all reviewers for their effort in reviewing our paper and in discussing their concerns with us during the rebuttal.

As there are misunderstandings in the initial comments, we tried to address all of them with our detailed responses below. Should there be any newly-raised concerns, we encourage the reviewers to share them with us. We are certain that we can address most/all of them through smooth communication.

---

### Author Response · Authors · 2022-11-18
**On understanding our big learning.**

Different from most machine learning (ML) papers focusing on a specific task, our big learning, potentially serving as a universal ML paradigm, covers a broad range of knowledge from many research fields, e.g., foundation models, generative modeling, adversarial learning, transformers/ViTs, NLP, etc. Therefore, misunderstandings from the audiences are expected and it indeed takes time and communication to fully understand what we are trying to convey with the presented big learning.

To clear up misunderstandings, we first address the common concerns raised by some reviewers, followed by carefully answering the questions from each reviewer.

We also encourage the reviewers to challenge us with their new questions if any. Please share with us your concerns, which we’d like to address in detail with confidence. Thanks.

---

### Author Response · Authors · 2022-11-18
**On comparisons with existing methods**

First of all, there are NO such existing methods that are capable of simultaneously modeling many/all
joint/conditional/marginal data distributions with complete/incomplete data, to our knowledge. Accordingly, we consider
both proposing the big learning with those capabilities and verifying its feasibility as solid contributions.

Secondly, on comparisons with existing heuristic learning methods for foundation models (like the mask-and-predict), we
emphasized several times in our paper that our big learning contains such heuristic methods as special cases (please see
the detailed discussions in the first 3 paragraphs of Page 5). We consider revealing the principled intuitively-simple
big learning framework to unify those heuristic methods as one of our contributions. We further contribute by revealing
that our big learning comes with many advantages, e.g., it has extraordinary data flexibilities that enable learning
with complete/incomplete data simultaneously. No existing learning methods for foundation models deliver those
advantages.

Finally, to alleviate the concerns about empirical comparisons, we conduct new experiments on the GLUE datasets [1] and
reveal that our big learning is a superior fine-tuning strategy to the naïve one. Specifically, we employ the
pretrained `xlnet-base-cased` model provided by the Hugging Face transformers library, followed by continually training
it on the downstream RTE/MRPC/SST-2 tasks with

1. naïve fine-tuning (i.e., identical to the XLNET that outperforms the BERT on many NLP tasks; termed FT) and
2. our big learning (termed big-learn), respectively.

Table 1 shows the tested hyperparameters by referring to the BERT and XLNET papers. The main empirical
results are summarized in Table 2 (see Appendix G of the revised manuscript for details), where it’s clear that our
big-learn consistently outperforms the naïve finetuning with boosted performance.

Table 1. Tested hyperparameters when comparing FT with big-learn.

| Task \ Hyperparameter | Learning Rate           | # Epochs            | WarmUp Steps | $\beta_{\text{BigLearn}}$ |
|-----------------------|-------------------------|---------------------|--------------|---------------------------|
| RTE                   | [2e-5, 4e-5, 6e-5]      | [3, 4, 7, 10, 15]   | [0, 120]     | [0., 0.2, 0.4, 0.6, 0.8]  |
| MRPC                  | [2e-5, 4e-5, 6e-5]      | [3, 4, 7, 10, 15]   | [0, 120]     | [0., 0.2, 0.4, 0.6, 0.8]  |
| SST-2                 | [2e-5, 4e-5, 6e-5]      | [2, 3, 4]           | [0, 1200]    | [0., 0.2, 0.4]            |

Table 2. Empirical evaluations showing the superiority of big-learn to FT. The best/median metrics are calculated among
the combinations of the tested hyperparameters in Table 1.

| Task \ Metric | Best Acc/F1 | Best Acc/F1      | Median-Acc / IQR | Median-Acc / IQR |
|---------------|-------------|------------------|------------------|------------------|
|               | FT          | big-learn        | FT               | big-learn        |
| RTE           | 71.84       | $\textbf{75.09}$ | 66.06/2.34       | $\textbf{70.75/1.44}$       |
| MRPC          | 88.97/92.09 | $\textbf{90.20/93.03}$      | 87.00/2.45       | $\textbf{87.74/1.10}$       |
| SST-2         | 94.15       | $\textbf{95.18}$            | 93.75/0.45       | $\textbf{94.66/0.28}$       |

Note intuitively, one would expect using our big learning to do the pretraining (in place of the permutation language
modeling of the XLNET), followed by applying the same naïve fine-tuning on downstream tasks, to evaluate the
effectiveness of our big learning. However, we cannot afford the pretraining cost, which takes about `5.5 days` on
`512 TPUs` to pretrain a XLNet-Large according to the XLNET paper; we leave that to the community, as mentioned in the
Conclusion. Alternatively, we reveal that big learning can serve as a superior fine-tuning strategy.

Theoretically, big learning can reduce the pretrain-finetuning gap, because it can act as the pretraining and finetuning
objectives simultaneously. Therefore, thanks also to its great data/task flexibilities, it’s quite
likely that the proposed big learning can also serve as a superior pretraining strategy to existing ones. Please refer
to the performance boost from BERT to XLNET and our discussions “on the generalization of model parameters and latent features” in Section 3.2.

[1] A. Wang, A. Singh, J. Michael, F. Hill, O. Levy, and S. Bowman. GLUE: A multi-task benchmark and analysis platform
for natural language understanding. In ICLR, 2018

---

### Author Response · Authors · 2022-11-18
**On the (downstream) use/value of big learning and big-learned capabilities**

Please see the second item of our contributions in the Introduction for the valuable “use” of the
joint/conditional/marginal data capabilities provided by our big learning. To be more specific,

- On the one hand, these data capabilities (big learned with complete/incomplete data) are directly associated with a
  tremendous number of valuable applications, such
  as arbitrary recommendation/completion, diverse data augmentation, flexible counterfactual analysis/reasoning, etc.
- On the other hand, Table 2 of our responses “On comparisons with existing methods” empirically proves that our big
  learning
  can serve as a superior fine-tuning strategy.

Because of the expensive pretraining cost, we cannot empirically verify that
our big learning is also a superior pretraining strategy to existing ones, even though that's probably the case (refer
to its successful special cases of BERT/XLNET/MAE, the performance boost from BERT to XLNET, and our discussions “on the
generalization of model parameters and
latent features” in Section 3.2). We share our thoughts with the community so that someone may help with that.

---

### Decision · Program_Chairs · 2023-01-20

**Decision:**

Reject

**Justification For Why Not Higher Score:**

The paper definitely needs to be rewritten with toned-down claims and needs to go through a detailed review process once again.

**Justification For Why Not Lower Score:**

N/A

**Metareview: Summary, Strengths And Weaknesses:**

This paper introduces a new framework called "big learning" which unifies the training of large-scale transformer models. The authors show that most of the commonly used objectives are special cases of big learning and that big learning can model the joint distribution better.

However, all three reviewers think that the claims in the paper are overstated and are not clear. This paper would benefit from a complete revision based on the reviewers' feedback. While the authors made a revision, the revision is not highlighted by a colour change and hence it is not easy for the reviewers to read the whole paper once again.

I recommend the authors work on the claims and writing and resubmit to a future venue. Also, I find the tone of the authors to be a bit rude towards the reviewers. While I understand that it is frustrating that the reviewers did not get what you intend to convey, it also shows that more work needs to be done from the authors' side in terms of explaining your ideas clearly.